# SARS-CoV-2 Antibody Testing in Health Care Workers: A Comparison of the Clinical Performance of Three Commercially Available Antibody Assays

Niamh Allen,[a] Melissa Brady,[b,c] Antonio Isidro Carrion Martin,[d] Lisa Domegan,[b] Cathal Walsh,[b,e,f] Elaine Houlihan,[g] Colm Kerr,[a] Lorraine Doherty,[b] Joanne King,[h] Martina Doheny,[i] Damian Griffin,[i] Maria Molloy,[j] Jean Dunne,[k] Vivion Crowley,[l] Philip Holmes,[l] Evan Keogh,[l] Sean Naughton,[l] Martina Kelly,[m] Fiona O'Rourke,[m] Yvonne Lynagh,[m] Brendan Crowley,[m] Cillian de Gascun,[n] Paul Holder,[n] Colm Bergin,[a,o] Catherine Fleming,[p] Una Ni Riain,[g] Niall Conlon,[k,o] on behalf of the PRECISE Study Steering Group

[a]Department of GU Medicine and Infectious Diseases, St. James's Hospital, Dublin, Ireland
[b]Health Service Executive-Health Protection Surveillance Centre (HPSC), Dublin, Ireland
[c]European Programme for Intervention Epidemiology Training (EPIET), European Centre for Disease Prevention and Control (ECDC), Stockholm, Sweden
[d]Department of Public Health, University of Murcia, Murcia, Spain
[e]Health Research Institute and MACSI, University of Limerick, Limerick, Ireland
[f]MISA and NCPE, St James's Hospital, Dublin, Ireland
[g]Department of Microbiology, University Hospital Galway, Galway, Ireland
[h]Department of Virology, University Hospital Galway, Galway, Ireland
[i]Department of Clinical Biochemistry, University Hospital Galway, Galway, Ireland
[j]Galway University Hospital Group, Galway, Ireland
[k]Department of Immunology, St. James's Hospital, Dublin, Ireland
[l]Department of Biochemistry, St. James's Hospital, Dublin, Ireland
[m]Department of Virology, St. James's Hospital, Dublin, Ireland
[n]National Virus Reference Laboratory, Dublin, Ireland
[o]Department of Clinical Medicine, Trinity College, Dublin, Ireland
[p]Department of Infectious Diseases, University Hospital Galway, Galway, Ireland

Niamh Allen and Melissa Brady contributed equally to this article. Niamh Allen was named first as she was the principal investigator for the study, she instigated and organized this research, and she took the lead in coordinating the analysis and writing the manuscript.

**ABSTRACT** Severe acute respiratory syndrome coronavirus 2 (SARS-CoV-2) antibodies are an excellent indicator of past COVID-19 infection. As the COVID-19 pandemic progresses, retained sensitivity over time is an important quality in an antibody assay that is to be used for the purpose of population seroprevalence studies. We compared 5,788 health care worker (HCW) serum samples by using two serological assays (Abbott SARS-CoV-2 anti-nucleocapsid immunoglobulin G (IgG) and Roche anti-SARS-CoV-2 anti-nucleocapsid total antibody) and a subset of samples (all Abbott assay positive or grayzone, $n = 485$) on Wantai SARS-CoV-2 anti-spike antibody enzyme-linked immunosorbent assay (ELISA). For 367 samples from HCW with a previous PCR-confirmed SARS-CoV-2 infection, we correlated the timing of infection with assay results. Overall, seroprevalence was 4.2% on Abbott and 9.5% on Roche. Of those with previously confirmed infection, 41% (150/367) and 95% (348/367) tested positive on Abbott and Roche, respectively. At 21 weeks (150 days) after confirmed infection, positivity on Abbott started to decline. Roche positivity was retained for the entire study period (33 weeks). Factors associated ($P \leq 0.050$) with Abbott seronegativity in those with previous PCR-confirmed infection included sex (odds ratio [OR], 0.30 male ; 95% confidence interval [CI], 0.15 to 0.60), symptom severity (OR 0.19 severe symptoms; 95% CI, 0.05 to 0.61), ethnicity (OR, 0.28 Asian ethnicity; 95% CI, 0.12 to 0.60), and time since PCR diagnosis (OR, 2.06 for infection 6 months previously; 95% CI, 1.01 to 4.30). Wantai detected all previously confirmed infections. In our

Address correspondence to Niamh Allen, nallen@stjames.ie, or Melissa Brady, melissa.brady1@hse.ie.

Comparison of SARS-CoV-2 antibody assays in healthcare workers; superior performance of anti-N panantibody assays over anti-N IgG assay @stjamesdublin @hpscireland

population, Roche detected antibodies up to at least 7 months after natural infection with SARS-CoV-2. This finding indicates that the Roche total antibody assay is better suited than Abbott IgG assay to population-based studies. Wantai demonstrated high sensitivity, but sample selection was biased. The relationship between serological response and functional immunity to SARS-CoV-2 infection needs to be delineated.

**IMPORTANCE** As the COVID-19 pandemic progresses, retained sensitivity over time is an important quality in an antibody assay that is to be used for the purpose of population seroprevalence studies. There is a relative paucity of published literature in this field to help guide public health specialists when planning seroprevalence studies. In this study, we compared results of 5,788 health care worker blood samples tested by using two assays (Roche and Elecsys, anti-nucleocapsid antibody) and by testing a subset on a third assay (Wantai enzyme-linked immunosorbent assay [ELISA] anti-spike antibody). We found significant differences in the performance of these assays, especially with distance in time from PCR-confirmed COVID-19 infection, and we feel these results may significantly impact the choice of assay for others conducting similar studies.

**KEYWORDS** Abbott Architect SARS-CoV-2 IgG, COVID-19 serological assays, Roche Elecsys SARS-CoV-2 panantibody, SARS-CoV-2 serological assay, SARS-CoV-2 seroprevalence, antibody assays SARS-CoV-2

Serological assays for the detection of severe acute respiratory syndrome coronavirus 2 (SARS-CoV-2) antibodies have a significant role to play in the response to the COVID-19 pandemic. The detection of antibodies to SARS-CoV-2 is an excellent indicator of past SARS-CoV-2 infection (1) and therefore helps determine the proportion of a population that has been previously exposed or infected. Population seroprevalence studies can help account for asymptomatic cases and symptomatic individuals who may not present to health services for testing. As mass COVID-19 vaccination programs are rolled out globally, serological assays may have a role in assessing host vaccine response and predicting vaccine effectiveness (2). They may also potentially be used to inform individual risk of disease (3) with emerging evidence of up to 6 months of immunity after natural infection (4), although further research in this area of postinfection immunity is required.

Many studies have examined the timing of antibody (Ab) production and sensitivity of commercially available antibody assays in the early stages of infection with SARS-CoV-2 (5) (5). Few studies have evaluated and compared the longevity of assay sensitivity after PCR-confirmed infection for different assays on the same health care worker (HCW) population. As the pandemic progresses, retained sensitivity over time is an important quality in an antibody assay that is to be used for the purpose of population seroprevalence studies. A decline in SARS-CoV-2 antibody response to natural infection over time has been described (6, 7), and different antibodies may wane at different rates. Other studies have shown sustained antibody detection for up to 100 to 125 days (8, 9). Understanding the duration of antibody response over time is key to understanding the accuracy of epidemiological studies and informing public health pandemic response measures. The extent and duration of immunity and its relationship to antibody positivity are not yet fully understood; however, the use of serology at an individual level for the purpose of estimating immunity also requires detailed understanding of the timing and waning of antibody response in relation to PCR positivity. A UK study comparing five widely available commercial assays (including the Abbott Architect immunoglobulin G (IgG) assay and Roche Elecsys anti-SARS-CoV-2 assay) found them all to have a sensitivity and specificity of at least 98% at 30 days after symptom onset (10). Regarding the longevity of the antibody response, there are few published data, and results differ; a study comparing four different serological assays showed a decline in the performance of the Abbott assay after 60 days and a further decline after 80 days (11), whereas another study of the same assay showed a

mean of 137 days to loss of positive antibody (12). A study comparing the same assays as our study highlighted better performance of the Roche assay over the Abbott assay, although numbers were small (13). Further data comparing antibody assays for the detection of antibodies to SARS-CoV-2 are needed to guide the most appropriate use of certain commercially available assays in any given situation.

The primary purpose of the data collected was to estimate the seroprevalence of past SARS-CoV-2 infection in our HCW population; that analysis was carried separately as part of the PRECISE Study on Prevalence of Antibodies to SARS-CoV-2 in Irish Healthcare Workers (14). The data presented here are a secondary analysis as part of the PRECISE study, with the aim of comparing the prevalence of anti-SARS-CoV-2 antibodies in the same HCW population in two different assays targeting the anti-nucleocapsid protein, namely, Abbott Architect SARS-CoV-2 IgG assay and Roche Elecsys anti-SARS-CoV-2 immunoassay (15–17). We aimed to also compare antibody prevalence for a subset of samples on an assay targeting the anti-spike protein, namely, the Wantai SARS-CoV-2 antibody ELISA.

## RESULTS

COVID-19 serology test results were available for 5,788 participating HCWs (comprising 64% of all staff in 2 Irish tertiary referral hospitals). The majority of participants were female (77%); the median age was 39 years (interquartile range [IQR], 30 to 49). The characteristics of participants tested by serology assay are shown in Appendix 1A and 1B.

**SARS-CoV-2 seropositivity in relation to assay type: Abbott SARS-CoV-2 and Roche Anti-SARS-CoV-2.** All 5,788 participants were tested using the Abbott SARS-CoV-2 IgG assay and 99.9% ($n = 5,787$) were tested using the Roche Anti-SARS-CoV-2 assay. A considerably lower proportion of participants had a positive antibody result on the Abbott assay (4.2%) than the proportion that had a positive antibody result on the Roche assay (9.5%) (Appendix 2A).

Assay concordance was moderate (k, 0.53; 95% confidence interval [CI], 0.48 to 0.57) (Table 1). McNemar's test for the difference in proportions indicated a systematic difference in the proportion of positive results between the two assays ($P < 0.001$). Twenty-four participants tested positive on Abbott but negative on Roche. Of the 24 participants, 3 also tested positive on the Wantai assay, suggesting possible false-negative results on Roche for these 3 participants. Of the remaining 21 participants, all tested negative on Wantai; none had a prior PCR-confirmed SARS-CoV-2 infection. An analysis was carried out in order to explore the association between participant characteristics and discordant results between the Abbott and Roche assays, and there was no significant association observed (data not presented here).

**Abbott SARS-CoV-2 "Grayzone" results.** Four percentage of participants ($n = 221/5,787$) had results in the Abbott SARS-CoV-2 IgG grayzone. Of those participants, 67% ($n = 149/221$) tested positive on the Roche assay. Among all of those with grayzone results, 42% ($n = 93/221$) were previously diagnosed with COVID-19 (SARS-CoV-2 positive by PCR). There were no participant characteristics that were significantly associated with having a result in the Abbott grayzone. In order to explore whether the numerical value within the grayzone sample to calibrator (S/C) index range could be used to assist in the interpretation of the grayzone, arbitrary cutoffs (low, medium, and

**TABLE 1** Distribution of positive and negative results by serology assay

| Roche anti-SARS-CoV-2 result | Abbott SARS-CoV-2 IgG ($n$) | | | Percentage agreement | $\kappa^a$ Statistic (95% CI) |
| | Positive result | Negative or grayzone result | Total | | |
| --- | --- | --- | --- | --- | --- |
| Positive | 218 | 329 | 547 | 93.9 | 0.53 (0.48–0.57) |
| Negative | 24 | 5,216 | 5,240 | | |
| Total | 242 | 5545 | 5,787 | | |

$^a\kappa$, Cohen's kappa.

**TABLE 2** Distribution of positive and negative results by assay

| Roche anti-SARS-CoV-2 result | Wantai SARS-CoV-2 Ab ELISA (n) | | | Percentage agreement | κ[a] Statistic (95% CI) |
|---|---|---|---|---|---|
| | Positive result | Negative result | Total | | |
| Positive | 386 | 3 | 389 | 97.7 | 0.93 (0.88–0.97) |
| Negative | 8 | 88 | 96 | | |
| Total | 394 | 91 | 485 | | |

[a]κ, Cohen's kappa.

high) were applied, and they were compared to the interpreted results on the Roche assay. There was no correlation observed, as a similar proportion of results within each of the arbitrary cutoffs was positive on the Roche assay (Appendix 2B).

**SARS-CoV-2 seropositivity in relation to assay type: Roche Anti-SARS-CoV-2 and Wantai SARS-CoV-2 Ab ELISA.** In total, 8.4% (n = 485/5,788) of participants were tested using the Wantai assay (of whom all were also tested using the Roche assay). Among these 485 participants, assay concordance was almost perfect (k, 0.93; 95% CI, 0.88 to 0.97) (Table 2). There was no evidence of differences in the proportion of positive results between the two assays (P = 0.131).

**SARS-CoV-2 seropositivity over time (previously PCR-positive participants).** A total of 367 participants were previously diagnosed COVID-19 positive by PCR. Among those participants, 41% (150/367) (95% CI, 35 to 46) tested positive on Abbott and 95% (348/367) (95% CI, 92 to 96) tested positive on Roche. There were 93 participants who were previously PCR positive who had a result in the Abbott grayzone. If the Abbott grayzone was included in positive results, Abbott positivity would increase to 66% (243/367). Of those participants diagnosed previously as COVID-19 positive by PCR, 71% (n = 259/367) were tested on Wantai, and all 259 had a positive result on the Wantai assay. Results for the Wantai assay are excluded from this section, as not all samples were tested on this platform.

The self-reported date of a previous positive PCR test was available for 365 (99%) participants. The interval between date of previous positive PCR test and date of serology test ranged from 12 to 231 days (2 to 33 weeks; 0 to 7 months). Serology test results by the number of weeks since a positive PCR test (including the breakdown for those who were symptomatic at the time of PCR testing) are shown in Table 3 and visually represented (by number of days) in Fig. 1 (visual representation excludes participants with grayzone results). We saw a decline in antibody positivity on the Abbott assay from week 21 (day 150) onward. The percent positivity by the number of months since a PCR test, including 95% CIs, is visually represented in Appendix 3A and 3B and shows a decline in antibody positivity on the Abbott assay in month 4.

The most common interval between positive PCR test and serology testing was 6 months (61%; n = 222). There were 210 participants who had a 6-month PCR-to-serology testing interval and who were symptomatic at the time of their PCR test; among them, positivity was 35% (95% CI, 29 to 41) on Abbott (n = 73) and was 93% (95% CI, 89 to 96) on Roche (n = 196).

Seventeen participants (4.6%) with a previously PCR-confirmed infection had a negative serology test result on both the Abbott and Roche assays and also tested negative on the Wantai assay. Of these 17 participants, 1 had a recent COVID-19 infection (24 days prior) and the remaining 16 had a distant infection (≥5 months prior to serology testing). Characteristics of the 17 participants are shown in Appendix 3C. Among the 17 participants, a lower proportion (88%; n = 15) had symptoms at the time of their PCR-positive test than the overall PCR-positive subgroup (98%; n = 358). A higher proportion participants were of white Irish background (94% versus 65% in the overall PCR-positive subgroup). Other characteristics did not differ considerably. An analysis was carried out in order to explore the association between participant characteristics and negative serology results; there was no significant association observed (data not presented here).

**TABLE 3** Detection of SARS-CoV-2 antibodies by serology assay type with respect to time since positive PCR test[a]

| No. of wks | Previously PCR positive (n = 367) | | | | | | | | | | Previously PCR positive and had symptoms at time of PCR test (n = 340) | | | | | | | | | |
| | Abbott SARS-CoV-2 IgG | | | | | | Roche anti-SARS-CoV-2 | | | | Abbott SARS-CoV-2 IgG | | | | | | Roche anti-SARS-CoV-2 | | | |
| | N | G | P | T | % Positive | % Positive (valid test) | N | P | T | % Positive | N | G | P | T | % Positive | % Positive (valid test) | N | P | T | % Positive |
|---|---|---|---|---|---|---|---|---|---|---|---|---|---|---|---|---|---|---|---|---|
| 2 | 0 | 0 | 5 | 5 | 100 | 100 | 1 | 4 | 5 | 80.0 | 0 | 0 | 2 | 2 | 100 | 100 | 1 | 1 | 2 | 50.0 |
| 3 | 1 | 0 | 0 | 1 | 0.0 | 0.0 | 1 | 0 | 1 | 0.0 | 1 | 0 | 0 | 1 | 0.0 | 0.0 | 1 | 0 | 1 | 0.0 |
| 4 | 0 | 1 | 2 | 3 | 66.7 | 100 | 1 | 2 | 3 | 66.7 | 0 | 1 | 2 | 3 | 66.7 | 100 | 1 | 2 | 3 | 66.7 |
| 5 | 0 | 0 | 4 | 4 | 100 | 100 | 0 | 4 | 4 | 100 | 0 | 0 | 4 | 4 | 100 | 100 | 0 | 4 | 4 | 100 |
| 6 | 0 | 0 | 1 | 1 | 100 | 100 | 0 | 1 | 1 | 100 | 0 | 0 | 1 | 1 | 100 | 100 | 0 | 1 | 1 | 100 |
| 7 | 0 | 0 | 2 | 2 | 100 | 100 | 0 | 2 | 2 | 100 | 0 | 0 | 2 | 2 | 100 | 100 | 0 | 2 | 2 | 100 |
| 9 | 0 | 0 | 1 | 1 | 100 | 100 | 0 | 1 | 1 | 100 | 0 | 0 | 1 | 1 | 100 | 100 | 0 | 1 | 1 | 100 |
| 10 | 0 | 0 | 2 | 2 | 100 | 100 | 0 | 2 | 2 | 100 | 0 | 0 | 2 | 2 | 100 | 100 | 0 | 2 | 2 | 100 |
| 11 | 0 | 0 | 1 | 1 | 100 | 100 | 0 | 1 | 1 | 100 | 0 | 0 | 0 | 0 | 100 | 100 | 0 | 0 | 0 | 100 |
| 17 | 0 | 0 | 1 | 1 | 100 | 100 | 0 | 1 | 1 | 100 | 0 | 0 | 1 | 1 | 100 | 100 | 0 | 1 | 1 | 100 |
| 19 | 0 | 0 | 2 | 2 | 100 | 100 | 0 | 2 | 2 | 100 | 0 | 0 | 2 | 2 | 100 | 100 | 0 | 2 | 2 | 100 |
| 20 | 0 | 0 | 1 | 1 | 100 | 100 | 0 | 1 | 1 | 100 | 0 | 0 | 1 | 1 | 100 | 100 | 0 | 1 | 1 | 100 |
| 21 | 0 | 1 | 5 | 6 | 83.3 | 100 | 0 | 6 | 6 | 100 | 0 | 1 | 5 | 6 | 83.3 | 100 | 0 | 6 | 6 | 100 |
| 22 | 0 | 4 | 7 | 11 | 63.6 | 100 | 0 | 11 | 11 | 100 | 0 | 3 | 6 | 9 | 66.7 | 100 | 0 | 9 | 9 | 100 |
| 23 | 3 | 1 | 7 | 11 | 63.6 | 70.0 | 1 | 10 | 11 | 90.9 | 2 | 1 | 7 | 10 | 70.0 | 77.8 | 0 | 10 | 10 | 100 |
| 24 | 6 | 5 | 6 | 17 | 35.3 | 50.0 | 0 | 17 | 17 | 100 | 5 | 5 | 5 | 15 | 33.3 | 50.0 | 0 | 15 | 15 | 100 |
| 25 | 9 | 12 | 12 | 33 | 36.4 | 57.1 | 0 | 33 | 33 | 100 | 9 | 11 | 12 | 32 | 37.5 | 57.1 | 0 | 32 | 32 | 100 |
| 26 | 7 | 10 | 7 | 24 | 29.2 | 50.0 | 1 | 23 | 24 | 95.8 | 7 | 9 | 7 | 23 | 30.4 | 50.0 | 1 | 22 | 23 | 95.7 |
| 27 | 18 | 7 | 23 | 48 | 47.9 | 56.1 | 2 | 46 | 48 | 95.8 | 16 | 7 | 21 | 44 | 47.7 | 56.8 | 2 | 42 | 44 | 95.5 |
| 28 | 20 | 9 | 18 | 47 | 38.3 | 47.4 | 2 | 45 | 47 | 95.7 | 19 | 8 | 17 | 44 | 38.6 | 47.2 | 1 | 43 | 44 | 97.7 |
| 29 | 20 | 17 | 24 | 61 | 39.3 | 54.5 | 6 | 55 | 61 | 90.2 | 19 | 16 | 24 | 59 | 40.7 | 55.8 | 6 | 53 | 59 | 89.8 |
| 30 | 31 | 18 | 9 | 58 | 15.5 | 22.5 | 4 | 54 | 58 | 93.1 | 29 | 18 | 8 | 55 | 14.5 | 21.6 | 4 | 51 | 55 | 92.7 |
| 31 | 8 | 7 | 6 | 21 | 28.6 | 42.9 | 0 | 21 | 21 | 100 | 6 | 7 | 6 | 19 | 31.6 | 50.0 | 0 | 19 | 19 | 100 |
| 32 | 0 | 1 | 2 | 3 | 66.7 | 100 | 0 | 3 | 3 | 100 | 0 | 1 | 2 | 3 | 66.7 | 100 | 0 | 3 | 3 | 100 |
| 33 | 1 | 0 | 0 | 1 | 0.0 | 0.0 | 0 | 1 | 1 | 100 | 1 | 0 | 0 | 1 | 0.0 | 0.0 | 0 | 1 | 1 | 100 |
| Total | 124 | 93 | 150 | 367 | 40.9 | 54.7 | 19 | 348 | 367 | 94.8 | 114 | 88 | 138 | 340 | 40.6 | 54.8 | 17 | 323 | 340 | 95.0 |

[a]Excludes two participants for which time since positive PCR test was unknown. Values are serology test results, as follows: N, number negative; G, number grayzone; P, number positive; T, total tested; % positive valid test was unknown. Values are serology test results, as follows: N, number negative; G, number grayzone; P, number positive; T, total tested; % positive valid test excluding grayzone results.

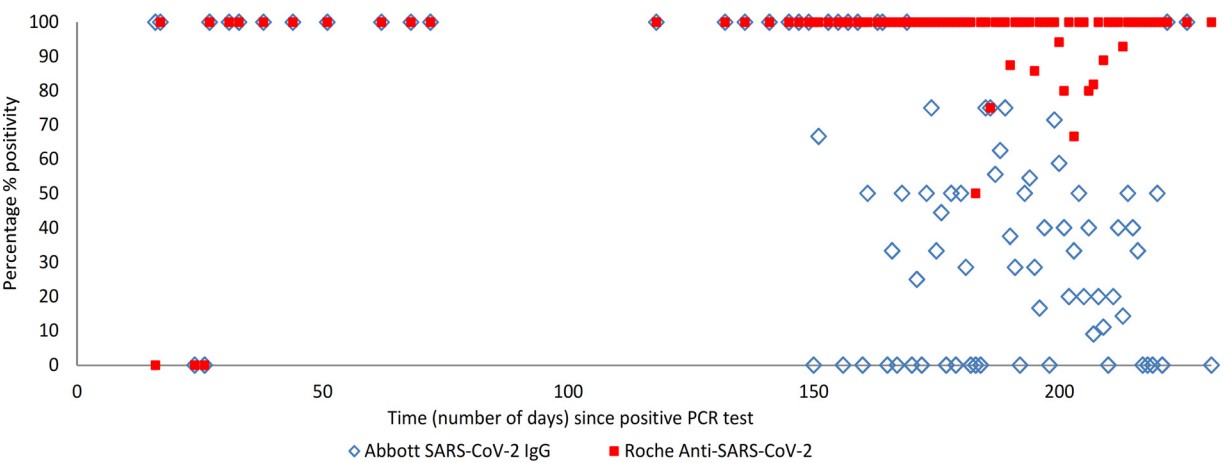

**FIG 1** Percentage of participants with detected SARS-CoV-2 antibodies by serology assay with respect to time (number of days) since a positive PCR test, among participants who had previous PCR-confirmed COVID-19 infection and had symptoms at the time of their PCR test (*n* = 340).

In order to explore whether the Roche quantitative cutoff index (COI) or the Abbott S/C index was close to the positive threshold for these 17 participants, arbitrary cutoffs (low, medium, and high negative) were applied to the results of both assays. For the Roche assay, three participants had results that were close to the positive result threshold (i.e., in the high negative range; COI, 0.6 to 0.9), six had results in the medium negative range (COI, 0.3 to 0.6), and eight had results in the low negative range (COI, 0 to 0.3). For the Abbott assay, all participants had results in either the medium negative range (*n* = 3) or the low negative range (*n* = 14).

**Abbott SARS-CoV-2 seronegativity in relation to participant characteristics.** The characteristics of participants with previous PCR-confirmed SARS-CoV-2 infection (*n* = 367) and their serology test results by Abbott SARS-CoV-2 are shown in Appendix 3D and 3E. In total, 45% (*n* = 124) (34% including those who had grayzone results in the total number tested) of participants with previous PCR-confirmed SARS-CoV-2 infection did not have detectable antibodies on the Abbott assay. Univariate and multivariable analyses were carried to out to explore the association between participant characteristics and the negative Abbott test result. Table 4 presents the results of the analysis. Factors associated with Abbott seronegativity in those with a previous PCR-confirmed infection varied by sex (male adjusted odds ratio [aOR], 0.30; 95% CI, 0.15 to 0.60; *P* < 0.001), symptom severity (severe symptoms requiring hospitalization; aOR, 0.19; 95% CI, 0.05 to 0.61; *P* = 0.008), ethnicity (Asian ethnicity; aOR, 0.28; 95% CI, 0.12 to 0.60; *P* = 0.001), and time since PCR diagnosis (increase from 5- to 6-month interval; aOR, 2.06; 95% CI, 1.01 to 4.30; *P* = 0.050).

To broaden the analysis, a multivariable logistic regression was repeated, including participants that had grayzone results. The results of this analysis were similar to the results of the initial analysis and are presented in Appendix 4. A separate analysis was carried out to explore the association between participant characteristics and Abbott grayzone results; there was no significant association observed (data not presented).

## DISCUSSION

**Seropositivity in relation to assay type: Abbott versus Roche—all participants.** In agreement with recently published data (11), a considerably higher proportion of participants (more than double) had detectable antibodies on the Roche assay than on the Abbott assay. In terms of assay performance, according to the manufacturer's instructions, both assays performed with high sensitivity and high specificity. Applying the lower bounds of the confidence interval for specificity of the Abbott assay, a maximum false-negative rate of 4.2% (*n* = 15) could be expected; therefore, 109 out of 124 negative test results among those previously diagnosed with COVID-19 (positive by

**TABLE 4** Factors associated with Abbott SARS-CoV-2 seronegativity, among participants with previous PCR-confirmed SARS-CoV-2 infection[a]

| Parameter | All participants (n) | Negative result (n) | Negative % | Unadjusted | | | Adjusted | | |
|---|---|---|---|---|---|---|---|---|---|
| | | | | OR | 95% CI | P value[b] | OR | 95% CI | P value[b] |
| Total | 274 | 124 | 45.3 | | | | | | |
| Age group (in yrs) | | | | | | | | | |
| 18–29 | 79 | 41 | 51.9 | *[c] | | | * | | |
| 30–39 | 79 | 43 | 54.4 | 1.11 | 0.59–2.07 | 0.750 | 0.99 | 0.46–2.15 | 0.982 |
| 40–49 | 59 | 17 | 28.8 | **0.38** | **0.18–0.76** | **0.007** | **0.36** | **0.15–0.82** | **0.017** |
| 50–59 | 45 | 19 | 42.2 | 0.68 | 0.32–1.41 | 0.301 | 0.70 | 0.28–1.71 | 0.429 |
| 60+ | 12 | 4 | 33.3 | 0.46 | 0.12–1.60 | 0.238 | 0.25 | 0.05–1.04 | 0.065 |
| Sex | | | | | | | | | |
| Female | 201 | 102 | 50.7 | * | | | * | | |
| Male | 73 | 22 | 30.1 | **0.42** | **0.25–0.74** | **0.003** | **0.30** | **0.15–0.60** | **<0.001** |
| Ethnicity | | | | | | | | | |
| Irish (white) | 173 | 94 | 54.3 | * | | | * | | |
| Any other white background | 35 | 12 | 34.3 | 0.44 | 0.20–0.92 | 0.033 | 0.49 | 0.2–1.19 | 0.118 |
| Any Asian background | 56 | 14 | 25.0 | **0.28** | **0.14–0.54** | **<0.001** | **0.28** | **0.12–0.60** | **0.001** |
| Any African/black background | 7 | 2 | 28.6 | 0.34 | 0.04–1.61 | 0.200 | 0.47 | 0.05–3.33 | 0.449 |
| Other | 3 | 2 | 66.7 | 1.68 | 0.15–36.6 | 0.067 | 2.33 | 0.14–70.6 | 0.566 |
| Close contact with a case of COVID-19[d] | | | | | | | | | |
| Yes | 155 | 65 | 41.9 | 0.71 | 0.44–1.15 | 0.178 | 0.63 | 0.34–1.16 | 0.137 |
| No | 117 | 59 | 50.4 | * | | | * | | |
| Main type of patient contact[e] | | | | | | | | | |
| No patient contact | 46 | 24 | 52.2 | * | | | * | | |
| Daily contact non-COVID-19 patients | 162 | 74 | 45.7 | 0.77 | 0.39–1.49 | 0.437 | 0.79 | 0.34–1.80 | 0.577 |
| Daily contact known or suspected COVID-19 patients | 66 | 26 | 39.4 | 0.60 | 0.27–1.27 | 0.182 | 0.83 | 0.32–2.14 | 0.694 |
| Severity of symptoms[d] | | | | | | | | | |
| No symptoms | 6 | 3 | 50.0 | 0.82 | 0.15–4.61 | 0.816 | 0.86 | 0.12–6.29 | 0.876 |
| Minor symptoms | 113 | 62 | 54.9 | * | | | * | | |
| Significant symptoms | 130 | 55 | 42.3 | **0.60** | **0.36–1.00** | **0.051** | **0.50** | **0.27–0.92** | **0.028** |
| Severe symptoms (hospitalized) | 23 | 4 | 17.4 | **0.17** | **0.04–0.49** | **0.003** | **0.19** | **0.05–0.61** | **0.008** |
| HCW role | | | | | | | | | |
| Administration | 18 | 9 | 50.0 | * | | | Not included in the model | | |
| Allied health care | 34 | 19 | 55.9 | 1.27 | 0.40–4.03 | 0.686 | | | |
| General support | 10 | 5 | 50.0 | 1.00 | 0.21–4.81 | 1.00 | | | |
| Health care assistant | 24 | 5 | 20.8 | 0.26 | 0.06–0.98 | 0.053 | | | |
| Medical/dental | 62 | 29 | 46.8 | 0.88 | 0.30–2.54 | 0.809 | | | |
| Nursing/midwifery | 124 | 55 | 44.4 | 0.80 | 0.29–2.17 | 0.653 | | | |
| Other | 2 | 2 | 100.0 | NA[f] | NA | NA | | | |
| No. of mo since PCR-positive test[d] | | | | | | | | | |
| Less than 1 | 7 | 1 | 14.3 | 0.25 | 0.01–1.57 | 0.21 | 0.11 | 0.01–0.83 | 0.061 |
| 1 | 8 | 0 | 0.0 | NA | NA | NA | NA | NA | NA |
| 2 | 4 | 0 | 0.0 | NA | NA | NA | NA | NA | NA |
| 3 | 1 | 0 | 0.0 | NA | NA | NA | NA | NA | NA |
| 4 | 10 | 0 | 0.0 | NA | NA | NA | NA | NA | NA |
| 5 | 57 | 23 | 40.4 | * | | | * | | |
| 6 | 168 | 91 | 54.2 | **1.75** | **0.95–3.25** | **0.073** | **2.06** | **1.01–4.30** | **0.050** |
| 7 | 17 | 9 | 52.9 | 1.66 | 0.57–5.05 | 0.360 | 2.67 | 0.74–10.2 | 0.139 |

[a]n = 274 participants with valid results.
[b]P values were calculated using the chi-square test, results for significant associations are highlighted in bold.
[c]*, reference category.
[d]Excludes 2 unknowns.
[e]Participants were asked which type of patient contact describes most of their current work (excludes 5 unknowns).
[f]NA, insufficient number of participants in this category.

PCR) could not be explained by expected test performance; it is possible that SARS-CoV-2 IgG was absent among these individuals. Applying the lower bounds of the confidence interval for specificity of the Roche assay, a maximum false-negative rate of 0.3% ($n = 1$) could be expected; therefore, 18 out of 19 negative test results among those previously diagnosed with COVID-19 could not be explained by expected test performance. It is possible that some of these individuals did not mount a SARS-CoV-2 antibody response to their infection; however, in the published literature, the majority of patients have been shown to develop antibodies following natural infection (18–20). Therefore, it is likely that the main reason for lack of antibodies in this subset of participants was due to a waning antibody response that was not picked up on testing, particularly on the Abbott assay. Studies have shown that anti-nucleocapsid IgG antibodies may be less likely to develop than IgM antibodies; Zhao et al. showed that at day 15 postinfection, 100% had detected total Ab, 94.3% had IgM, and only 79.8% had IgG Ab (18). Peterson et al. and Kaufman et al. showed that IgG antibodies were not detected for between 6.3% and 9% of people in the first 2 weeks after PCR confirmation of infection (20, 21) and that this proportion rose to 14% by 99 to 121 days (22).

Lumley at al. demonstrated stable anti-spike IgG antibodies but waning anti-nucleocapsid IgG antibodies over time among HCWs in the United Kingdom health care service (12), as did Pelleau et al. (23). This result may in part explain the Abbott IgG assay's relative performance for detecting past infection, compared with the Roche total antibody assay. The total antibody approach used in the Roche Elecsys system results in improved and sustained sensitivity suitable for population seroprevalence studies.

Twenty-four participants had detectable antibodies on the Abbott assay but not on the Roche assay. Of those participants, three had detectable antibodies on the Wantai assay (one of whom had recent PCR-confirmed infection) and were presumably false negatives on Roche, possibly reflecting differences in test method and target between the Wantai and Roche assays.

**Does the Abbott grayzone add anything?** In October 2020, Abbott updated their guidance on their Architect SARS-CoV-2 IgG assay (Abbott Diagnostics Product Information Letter PI1060-2020) to include an optional editable "grayzone" with a S/C index range of 0.5 to 1.39 which they advise "must be interpreted by the clinician in the context of relevant clinical and laboratory information on the patient" (16). The grayzone accounts for a large proportion of the disagreement between the assays. The majority of grayzones were positive by Roche (67%) and Wantai (70%). However, interpretation of this Abbott grayzone result in a clinical setting would not be straightforward, and confirmatory testing would still be needed. There was no obvious correlation between increasing or decreasing Abbott grayzone S/C indices and the interpreted results from other assays, so while the grayzone may add value for individual serology testing (by indicating a need for additional testing), we feel that it does not add value in the setting of population-based serological studies. In an epidemiological study setting such as this, inclusion of all grayzone results as presumptive positives would lead to a significant overestimation of seroprevalence. Studies such as ours could be used to estimate the proportion of grayzone results likely to be positive on other testing platforms; however, the use of alternative assays with prolonged sensitivity over time would be superior if the primary purpose is to estimate infection ever. It is notable that while studies have shown good protection against reinfection over at least a 6-month period following PCR-confirmed COVID-19 infection (4), to the best of our knowledge, no studies have yet compared antibody assays in terms of functional immunity.

**Seropositivity in relation to assay type—Roche versus Wantai assay.** We found almost perfect agreement between the Roche and Wantai assays. A study comparing eight assays found these two assays to provide the highest sensitivities at 98% and 95%, respectively (24), which, in agreement with our findings, suggest that total antibody assays are better positioned for population-based serological testing. However, in our study, the assay concordance should be interpreted with caution, as the selection of participants for additional testing on Wantai ELISA was heavily biased; only participants who had a positive or grayzone Abbott SARS-CoV-2 IgG result were selected for testing by the Wantai assay. The small degree of discordance between these two

total antibody assays could be due to a number of reasons, including the protein targeted (anti-N versus anti-S) and the different expected performances according to manufacturer guidelines (25).

**Seropositivity over time (previously PCR-positive participants).** Of the subset of participants who had previous PCR-confirmed COVID-19 infection, the Roche assay performed better in terms of identifying these participants and 95% tested positive. Only one-half of these past infections were accurately identified by the Abbott assay. Harley et al. showed a much higher level of agreement between these two assays, although on a smaller number of samples than were tested in our study and at an earlier time point after PCR-confirmation of infection (26). Therefore, antibody waning may not yet have occurred to the degree that it had occurred in our study All previously PCR-positive participants in our study had detectable antibodies on the Wantai ELISA, but this subpopulation examination and its inherent selection bias limit strong conclusions from these data.

The majority of participants had a 6-month interval between confirmed infection and antibody testing in our study, which correlated with the time period between the peak of the first wave of the pandemic in Ireland and the antibody testing in our study. Although detection was slightly lower on both assays than the overall detection of previous infections occurring at any time, the Roche assay still performed significantly better than Abbott using the 6-month time frame, identifying 93% versus 46% of infections that occurred 6 months prior.

The decay in seropositivity on Abbott in our cohort started at 21 weeks (150 days) after confirmed infection, but the small numbers of infection per week prior to this time should be noted. This decay did not change with removal of participants who had no symptoms at the time of infection. A study comparing four different serological assays, including the Abbott and Roche assays, showed a decline in the performance of the Abbott assay after 60 days, whereas antibodies were still detected on the Roche assay after 80 days (11). In contrast, another study of the Abbott assay showed a mean of 137 days to the loss of positive antibodies (12). Kumar et al. showed a much shorter duration of antibody detection using the Roche assay after confirmed SARS-CoV-2 infection in a small cohort of Indian HCWs; seropositivity halved by day 30 to 42 and fell to 0 after 50 days (27), which may suggest an impact of host factors, such as ethnicity, on antibody production and waning. In a large US study, Kaufman et al. showed a decline in IgG antibodies (including on Abbott) to 74% by 2 months after confirmed infection (22). Due to the small numbers of infection per week prior to the 21-week time point in our study, it is possible that this decline in Abbott IgG positivity started at an earlier time point that was not picked up in our study. While we demonstrate good performance of the Roche anti-nucleocapsid assay at extended time points, performance of assays at time points even more distant from the infection remains obscure.

In a multivariable analysis, the risk of a negative result on the Abbott platform despite previous PCR-confirmed COVID-19 infection increased with the interval between infection and antibody testing, in keeping with antibody waning. Those who had moderate or severe symptoms were more likely to have retained IgG than those who had only mild symptoms. This result was consistent with the findings of other researchers who found that IgG was better sustained in persons reporting significant symptoms than those who had mild or no symptoms (24, 26, 27). Participants of male sex and Asian ethnicity were also more likely to have sustained Abbott assay-detected IgG. Kaufman et al. showed age and male sex to be associated with the probability of persistent IgG serology (22), as did a large Cochrane review conducted in April 2020 (28). Furthermore, Lumley at al. demonstrated that increasing age, Asian ethnicity, and prior self-reported symptoms were independently associated with higher maximum anti-nucleocapsid IgG levels (12). This result is in keeping with our findings on sex, ethnicity, and prior self-reported symptoms; however, we did not find any statistically significant correlation with age. The reason for this sustained IgG response in participants of male

sex is not yet clear but may be related to higher viral load, other indices of severity, or other unknown biological factors not included in this study.

Participants who had a 6-month PCR-to-serology testing interval were twice as likely to be IgG seronegative (Abbott) than participants who had a 5-month testing interval. Those who had a 7-month testing interval were also more likely to be IgG seronegative than those who had a 5-month testing interval, but results were not significant. Our findings are consistent with the findings of other researchers who have demonstrated a decline in IgG seropositivity over time, by using the Abbott and other IgG antibody assays (6, 23). Further studies may provide a better understanding of seropositivity and seroreversion over time, relating to specific assays or the type of immunoglobulin measured.

We did not have enough participants with very recent infection to assess the ability of each assay to detect early infection; however, of the 8 infections that occurred within 4 weeks of antibody testing, 8/9 were correctly identified by the Abbott assay and 6/9 were identified by the Roche assay. Higher numbers would be needed to compare these assays specifically in the early stages of infection.

Almost 5% of infections were not identified by either platform. The majority were distant infections (≥6 months ago), and therefore, waning immunity may explain the seronegativity. One was a recent infection (PCR positive 24 days prior to serological testing) in a participant who had severe symptoms; this infection may have been too recent to have detectable antibodies on either assay; however, four other participants with only minor symptoms and even more recent infections (PCR positive 12 to 17 days prior to serology) had antibodies detected on both Abbott and Roche assays. Overall, fewer of these participants with previous confirmed infection and negative serology on both platforms were symptomatic at the time of their infection; other studies have shown those without symptoms to be both less likely to develop antibodies and less likely to have persistent antibodies 8 weeks postinfection (26). There were no other participant characteristics that were significantly associated with negative serology, nor were these participants quantitative results (Abbott S/C and Roche COI index) close to the positive result threshold for either assay. Both assays performed below their expected sensitivity based on manufacturer guidelines, likely due to the time interval between infection and antibody testing. This shortcoming should be considered in serological studies, as well as when using antibody testing in the setting of clinical care. As the pandemic progresses, further studies may highlight the sensitivity of different assays and/or different antigenic targets with more accuracy in relation to the timing of testing.

**Limitations.** Our study has several limitations. First, as the main PRECISE seroprevalence study was not designed with the primary objective of comparing the serological assays, we did not provide a comparable assessment of specificity by testing of any prepandemic negative-control samples. Second, information on COVID-19 symptoms and PCR test results were self-reported and thus could be biased. The dates of PCR-confirmed infection (also self-reported) may be inaccurate, and participants may have been reinfected at a later date but unaware of it; given the timing of this study at 8 months after the first COVID-19 cases in Ireland and the evidence of 9 months of protection from reinfection following natural infection, it is unlikely that many, if any, participants in the study had been reinfected (29). Furthermore, the PCR cycle threshold ($C_T$) value which would have been a valuable addition to this study was not available. Other variables which would have been valuable to this study but were not available include participant comorbidities and host determinants of immune response; there may be individual pathophysiological factors that influence the immune response differently depending on the antigenic target. A small sample size for the 0- to 4-month PCR-to-serology test interval prevented a meaningful analysis of seropositivity and seronegativity at the early stages postinfection.

Our study focused on assays that have the SARS CoV-2 nucleocapsid as an antigenic target with only a subpopulation assessed using a spike antibody assay. Given emerging evidence of differences in antibody decay related to the antigenic target, a more complete

assessment would have been desirable. Such approaches, including parallel spike and nucleocapsid assessments, may become more relevant in the era of SARS CoV-2 vaccination. In addition, we did not address the neutralizing capacity of the antibodies measured, and therefore, conclusions about the functional consequences of the presence of such antibodies are limited.

**Conclusion and recommendations.** The Roche total antibody assay performed significantly better at identifying those who had ever had a confirmed COVID-19 infection than the Abbott IgG assay, although both assays were less sensitive than the manufacturer guidelines. Our study findings suggest that, of these two assays directed at anti-nucleocapsid antibodies, Roche is better suited to future population-based serological studies due to a maintained detection of total antibodies up to at least 7 months after natural infection with SARS-CoV-2. While maximum sensitivity is achieved using multiple testing platforms, this is not always feasible or cost-effective, and the overall seroprevalence results of our study would have been unchanged if only the Roche assay was used. While we demonstrate good performance of the Roche anti-nucleocapsid assay at extended time points, results of the comparison with other antigenic targets at time points even more distant from the infection remain obscure. The anti-spike assay (Wantai) performed very well on the subset of samples analyzed, but further studies are needed to show if it maintains its sensitivity compared with Roche on an unbiased selection of samples. The performance of both total antibody assays (Roche and Wantai) points toward the superiority of total antibody assays over IgG assays for serological studies.

The risk of a negative antibody result on the Abbott assay despite a previous PCR-confirmed SARS-CoV-2 infection increased with time since infection; the decline was noted at 21 weeks/150 days after a confirmed SARS-CoV-2 infection. Those participants of male sex and Asian ethnicity, and those with moderate to severe symptoms were more likely to retain IgG detection on the Abbott assay. The Abbott grayzone did not add anything that would eliminate the need for additional testing on an alternative assay. From our limited numbers of recent infections, it may be possible that the Abbott assay identified early infection quicker than the Roche assay, but our numbers were too small to confirm this hypothesis.

Our study adds to the growing literature on serological assays for population-based studies. Further data comparing SARS-CoV-2 antibody assays are needed to guide the most appropriate use of certain assays in any given situation, especially where the use of dual or multiple assays is not feasible or affordable. With the recent introduction of widespread vaccination, it is yet to be determined if measuring antibody response post vaccination is meaningful and cost-effective and, if so, which assays are superior. Further studies are also needed to delineate the relationship between serological response and functional immunity to SARS-CoV-2 infection, both following vaccination and natural infection. Assays with a prolonged sensitivity are likely to be more valuable as the pandemic continues, in particular anti-nucleocapsid antibody assays that will differentiate past natural infection from vaccine-induced immunity.

## MATERIALS AND METHODS

**Study design.** This was a cross-sectional study of the seroprevalence of circulating antibodies to SARS-CoV-2 in hospital HCW, performed in October 2020 (30). All staff members of two Irish tertiary referral hospitals were invited to participate in an online self-administered consent process and online questionnaire, followed by blood sampling for SARS-CoV-2 antibody testing. These participants included both patient-facing and nonpatient facing staff members. Information collected in the questionnaire included demographic information and information regarding previous COVID-19 symptoms, testing, and diagnosis. Full details of the study design, study locations, recruitment, and study participants can be found in the published PRECISE study report (30).

**Ethical approval.** Ethical approval was obtained from the National Research Ethics Committee for COVID-19 in Ireland (20-NREC-COV-101).

**Laboratory assays.** All samples were tested by two assays, namely, Abbott SARS-CoV-2 measuring IgG (referred to here as Abbott) and Roche Elecsys Anti-SARS-CoV-2 measuring total antibodies (referred to here as Roche) (15–17). The Abbott assay is a chemiluminescent microparticle immunoassay that detects IgG antibodies to the nucleocapsid protein of SARS-CoV-2. The Roche assay is an electrochemiluminescent immunoassay that detects antibodies (including IgG), also to the nucleocapsid protein. Assay

**TABLE 5** Summary of assay performance according to manufacturer specifications

| Assay | Sensitivity (% [95% CI]) | Specificity (% [95% CI]) |
|---|---|---|
| Abbott SARS-CoV-2 IgG | 100 (95.8–100) | 99.6 (99.0–99.9) |
| Roche Anti-SARS-CoV-2 | 100 (88.3–100) | 99.8 (99.7–99.9) |
| Wantai SARS-CoV-2 Antibody ELISA | 96.7 (83.3–99.4) | 97.5 (91.3–99.3) |

results were interpreted using the manufacturers' recommended assay specific thresholds. Assay threshold of $\geq$1.4 (sample to calibrator [S/C] index) for Abbott and $\geq$1.0 (cutoff index [COI]) for Roche were determined to be reactive and interpreted as antibody positive. The Abbott SARS-CoV-2 IgG grayzone is an additional assay threshold band for potential positivity, suggested by the manufacturer to increase assay sensitivity (Abbott Diagnostics Product Information Letter PI1060-2020) (16). The interpretation of Abbott S/C indices within this study is as follows: negative, $<$0.5; grayzone, 0.5 to $<$1.4; and positive, $\geq$1.4. All samples with an Abbott result of positive or grayzone were tested on a third assay in the National Virus Reference Laboratory (NVRL) using the Wantai SARS-CoV-2 antibody ELISA (referred to here as Wantai), distributed by Fortress Diagnostics. Wantai is an enzyme-linked immunosorbent assay (ELISA) for qualitative detection of total antibodies (including IgG and IgM) to the spike protein of SARS-CoV-2.

In terms of assay performance, according to the manufacturer's specifications, all three assays perform with high sensitivity and high specificity (see Table 5) (25).

**Statistical methods.** Assay concordance was assessed using Cohen's kappa statistic and McNemar's test for difference in proportions. Cohen's kappa coefficient ($\kappa$) measures the level of agreement between assays, taking into account the possibility of the agreement occurring by chance. The $\kappa$ statistic varies from 0 to 1, where 0 = agreement equivalent to chance, 0.1 to 0.20 = slight agreement, 0.21 to 0.40 = fair agreement, 0.41 to 0.60 = moderate agreement, 0.61 to 0.80 = substantial agreement, 0.81 to 0.99 = near perfect agreement, and 1 = perfect agreement. For analysis of assay concordance, results must be classified into the same number of mutually exclusive categories; therefore, Abbott negative and grayzone results were grouped. Confidence intervals for the proportion of participants that were seropositive were computed. Multivariable logistic regression was carried out to assess risk factors for the absence of SARS-CoV-2 IgG antibodies by controlling for age, sex, ethnicity/background, type of patient contact, severity of symptoms, and number of months since a PCR positive test. Forward stepwise selection was used, and the Akaike information criterion (AIC) was used to evaluate model efficiency; HCW role was excluded in the final model. We used R version 4.0.3 (R Foundation for Statistical Computing, Vienna, Austria) and OpenEpi software version 3.01 (Wilson Score) (31).

**Data availability.** The original data set for this research may identify individuals and therefore is not being routinely deposited in a data repository. However, the data set can be made available upon reasonable request to the principal investigator.

# APPENDIX

**Appendix 1A** Characteristics of participants tested, by serology assay

| | Participants tested by: | | | | | | | |
| | Abbott SARS-CoV-2 IgG | | Roche anti-SARS-CoV-2 | | Wantai SARS-CoV-2 antibody ELISA | | Total participants | |
| Parameter | n | % | n | % | n | % | n | % |
|---|---|---|---|---|---|---|---|---|
| Total participants tested | 5,788 | | 5,787 | | 485 | | 5,788 | |
| | | | | | | | | |
| Age group (yrs) | | | | | | | | |
| 18–29 | 1,350 | 23.3 | 1,350 | 23.3 | 139 | 28.7 | 1,350 | 23.3 |
| 30–39 | 1,617 | 27.9 | 1,617 | 27.9 | 139 | 28.7 | 1,617 | 27.9 |
| 40–49 | 1,516 | 26.2 | 1,515 | 26.2 | 111 | 22.9 | 1,516 | 26.2 |
| 50–59 | 1,001 | 17.3 | 1,001 | 17.3 | 68 | 14.0 | 1,001 | 17.3 |
| 60+ | 304 | 5.3 | 304 | 5.3 | 28 | 5.8 | 304 | 5.3 |
| | | | | | | | | |
| Sex | | | | | | | | |
| Female | 4,478 | 77.4 | 4,478 | 77.4 | 349 | 72.0 | 4,478 | 77.4 |
| Male | 1,309 | 22.6 | 1,308 | 22.6 | 136 | 28.0 | 1,309 | 22.6 |
| Unknown | 1 | 0.0 | 1 | 0.0 | 0 | 0.0 | 1 | 0.0 |
| | | | | | | | | |
| Ethnicity | | | | | | | | |
| Irish (white) | 4,444 | 76.8 | 4,444 | 76.8 | 313 | 64.5 | 4,444 | 76.8 |
| Any other white background | 552 | 9.5 | 551 | 9.5 | 54 | 11.1 | 552 | 9.5 |
| Any Asian background | 577 | 10.0 | 577 | 10.0 | 101 | 20.8 | 577 | 10.0 |
| Any African or black background | 113 | 2.0 | 113 | 2.0 | 13 | 2.7 | 113 | 2.0 |
| Other | 101 | 1.7 | 101 | 1.7 | 4 | 0.8 | 101 | 1.7 |
| Unknown | 1 | 0.0 | 1 | 0.0 | 0 | 0 | 1 | 0.0 |
| | | | | | | | | |
| HCW role | | | | | | | | |
| Nursing/midwifery | 2,064 | 35.7 | 2,064 | 35.7 | 230 | 47.4 | 2,064 | 35.7 |
| Allied health | 1,091 | 18.8 | 1,091 | 18.9 | 57 | 11.8 | 1,091 | 18.8 |
| Medical/dental | 983 | 17.0 | 982 | 17.0 | 93 | 19.2 | 983 | 17.0 |
| Admin | 803 | 13.9 | 803 | 13.9 | 33 | 6.8 | 803 | 13.9 |
| General support | 434 | 7.5 | 434 | 7.5 | 21 | 4.3 | 434 | 7.5 |
| Health care assistant | 286 | 4.9 | 286 | 4.9 | 45 | 9.3 | 286 | 4.9 |
| Other | 127 | 2.2 | 127 | 2.2 | 6 | 1.2 | 127 | 2.2 |

**Appendix 1B** COVID-19 related characteristics of participants tested, by serology assay

| Parameter | Abbott SARS-CoV-2 IgG | | Roche Anti-SARS-CoV-2 | | Wantai SARS-CoV-2 Antibody ELISA | | Total participants | |
|---|---|---|---|---|---|---|---|---|
| | n | % | n | % | n | % | n | % |
| Total participants tested | 5,788 | | 5,787 | | 485 | | 5,788 | |
| Close contact with a case of COVID-19 | | | | | | | | |
| Yes | 1,705 | 29.5 | 1,704 | 29.4 | 258 | 53.2 | 1,705 | 29.5 |
| No | 4,071 | 70.3 | 4,071 | 70.3 | 225 | 46.4 | 4,071 | 70.3 |
| Unknown | 12 | 0.2 | 12 | 0.2 | 2 | 0.4 | 12 | 0.2 |
| Main type of patient contact[a] | | | | | | | | |
| Daily contact with known/suspected COVID-19 patients | 903 | 15.6 | 902 | 15.6 | 109 | 22.5 | 903 | 15.6 |
| Daily contact with patients without COVID-19 | 3,245 | 56.1 | 3,245 | 56.1 | 299 | 61.6 | 3,245 | 56.1 |
| No patient contact | 1,635 | 28.2 | 1,635 | 28.3 | 77 | 15.9 | 1,635 | 28.2 |
| Unknown | 5 | 0.1 | 5 | 0.1 | 0 | 0.0 | 5 | 0.1 |
| Previous COVID-19 symptoms (ever) | | | | | | | | |
| No symptoms | 2,869 | 49.6 | 2,868 | 49.6 | 102 | 21.0 | 2,869 | 49.6 |
| Had symptoms | 2911 | 50.3 | 2,911 | 50.3 | 381 | 78.6 | 2,911 | 50.3 |
| Unknown | 8 | 0.1 | 8 | 0.1 | 2 | 0.4 | 8 | 0.1 |
| Severity of symptoms | | | | | | | | |
| Minor symptoms | 2,159 | 74.2 | 2,159 | 74.2 | 193 | 50.7 | 2,159 | 74.2 |
| Significant symptoms | 701 | 24.1 | 701 | 24.1 | 161 | 42.3 | 701 | 24.1 |
| Severe symptoms (hospitalized) | 51 | 1.8 | 51 | 1.8 | 27 | 7.1 | 51 | 1.8 |
| Previous COVID-19 PCR test | | | | | | | | |
| Yes | 2,779 | 48.0 | 2,778 | 48.0 | 380 | 78.4 | 2,779 | 48.0 |
| No | 3,003 | 51.9 | 3,003 | 51.9 | 105 | 21.6 | 3,003 | 51.9 |
| Unknown | 6 | 0.1 | 6 | 0.1 | 0 | 0.0 | 6 | 0.1 |
| Previous positive COVID-19 PCR test | | | | | | | | |
| Yes | 367 | 6.3 | 367 | 6.3 | 259 | 53.4 | 367 | 6.3 |
| No | 5,415 | 93.6 | 5,414 | 93.6 | 226 | 46.6 | 5,415 | 93.6 |
| Unknown | 6 | 0.1 | 6 | 0.1 | 0 | 0.0 | 6 | 0.1 |

[a]Participants were asked which one describes most of their current work.

**Appendix 2A** Serology result by assay[a,d]

| Assay | Negative | Grayzone | Positive | Total[b] | % Positive | Total valid[c] | % Positive (where valid) |
|---|---|---|---|---|---|---|---|
| Abbott SARS-CoV-2 IgG | 5,325 | 221 | 242 | 5,788 | 4.2 | 5,567 | 4.3 |
| Roche Anti-SARS-CoV-2 | 5,240 | | 547 | 5,787 | 9.5 | 5,787 | 9.5 |

[a]Values are no. of participants unless otherwise stated.
[b]Total, total tested.
[c]Total valid, total no. tested excluding grayzone test result.
[d]This table excludes the Wantai assay due to sampling bias.

**Appendix 2B** Comparison of Abbott SARS-CoV-2 IgG S/C index and Roche anti-SARS-CoV-2 interpreted result, among participants with grayzone results[a]

| | Result by assay: | | | |
|---|---|---|---|---|
| | Abbott SARS-CoV-2 Grayzone (n) | Roche anti-SARS-CoV-2 | | |
| Arbitrary cutoff | | Negative (n) | Positive (n) | Positivity (%) |
| Low grayzone (S/C 0.5–0.8) | 115 | 39 | 76 | 66.1 |
| Medium grayzone (S/C 0.8–1.1) | 62 | 18 | 44 | 71.0 |
| High grayzone (S/C 1.1–1.4) | 44 | 15 | 29 | 65.9 |
| Total | 221 | 72 | 149 | 67.4 |

[a]n = 221.

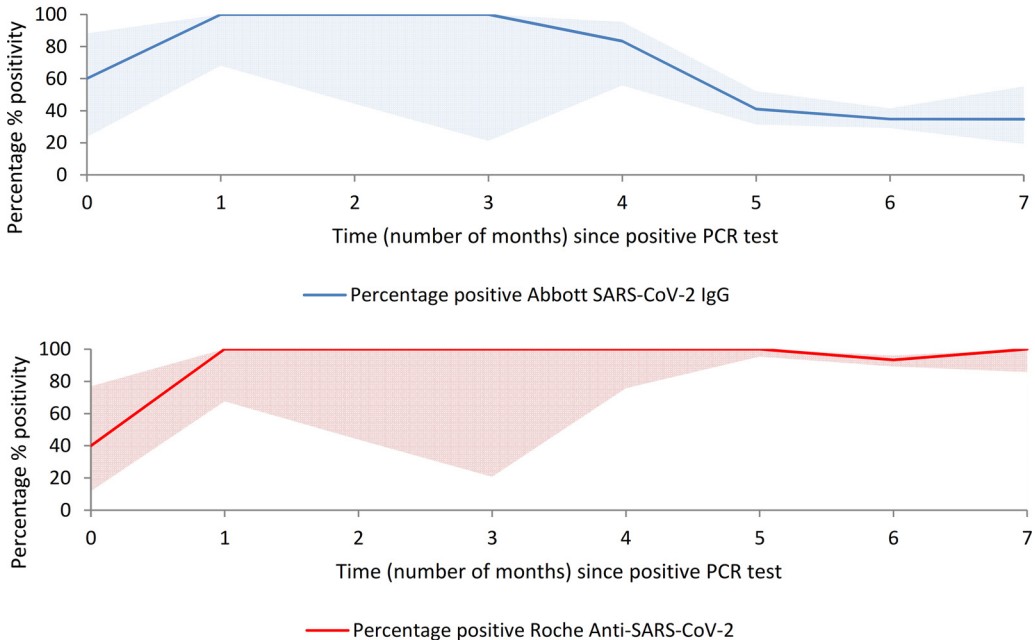

**Appendix 3A and 3B** Percentage of participants with detected SARS-CoV-2 antibodies by serology assay with respect to time (number of months) since a positive PCR test, among participants who had previous PCR-confirmed COVID-19 infection and had symptoms at the time of their PCR test ($n$ = 340). Shaded area shows the 95% confidence interval.

**Appendix 3C** Characteristics of participants who had a previously PCR-confirmed infection and were negative by serology testing

| Parameter | n | % |
|---|---|---|
| Total | 17 | |
| | | |
| Age group | | |
| 18–29 | 5 | 29.4 |
| 30–39 | 4 | 23.5 |
| 40–49 | 4 | 23.5 |
| 50–59 | 4 | 23.5 |
| 60+ | 0 | 0.0 |
| | | |
| Sex | | |
| Female | 14 | 82.4 |
| Male | 3 | 17.6 |
| | | |
| Ethnicity | | |
| Irish (white) | 16 | 94.1 |
| Any other white background | 1 | 5.9 |
| | | |
| HCW role | | |
| Nursing/midwifery | 7 | 41.2 |
| Medical/dental | 4 | 23.5 |
| Admin | 1 | 5.9 |
| Allied health | 2 | 11.8 |
| Health care assistant | 2 | 11.8 |
| General support | 1 | 5.9 |
| | | |
| Close contact with a case of COVID-19 | | |
| Yes | 7 | 41.2 |
| No | 10 | 58.8 |
| | | |
| Main type of contact with COVID-19 patients | | |
| Daily contact with known/suspected COVID-19 patients | 3 | 17.6 |
| Daily contact with patients without COVID-19 | 11 | 64.7 |

**Appendix 3C** (Continued)

| Parameter | *n* | % |
|---|---|---|
| No patient contact | 3 | 17.6 |
| | | |
| Previous COVID-19 symptoms | | |
| No symptoms | 2 | 11.8 |
| Had symptoms | 15 | 88.2 |
| | | |
| Severity of symptoms | | |
| Minor symptoms | 7 | 46.7 |
| Significant symptoms | 8 | 53.3 |
| Severe symptoms (hospitalized) | 0 | 0.0 |
| | | |
| No. of mo since positive PCR test | | |
| Less than 1 | 1 | 5.9 |
| 5 | 1 | 5.9 |
| 6 | 15 | 88.2 |

**Appendix 3D** Characteristics of participants who had a previously PCR-confirmed infection and their test results on Abbott SARS-CoV-2 IgG

| Parameter | All participants | | Test result | | | | | |
|---|---|---|---|---|---|---|---|---|
| | | | Grayzone | | Positive | | Negative | |
| | *n* | % | *n* | % | *n* | % | *n* | % |
| Total | 367 | | 93 | | 150 | | 124 | |
| | | | | | | | | |
| Age group | | | | | | | | |
| 18–29 | 111 | 30.2 | 32 | 34.4 | 38 | 25.3 | 41 | 33.1 |
| 30–39 | 108 | 29.4 | 29 | 31.2 | 36 | 24.0 | 43 | 34.7 |
| 40–49 | 79 | 21.5 | 20 | 21.5 | 42 | 28.0 | 17 | 13.7 |
| 50–59 | 50 | 13.6 | 5 | 5.4 | 26 | 17.3 | 19 | 15.3 |
| 60+ | 19 | 5.2 | 7 | 7.5 | 8 | 5.3 | 4 | 3.2 |
| | | | | | | | | |
| Sex | | | | | | | | |
| Female | 269 | 73.3 | 68 | 73.1 | 99 | 66.0 | 102 | 82.3 |
| Male | 98 | 26.7 | 25 | 26.9 | 51 | 34.0 | 22 | 17.7 |
| | | | | | | | | |
| Ethnicity | | | | | | | | |
| Irish (white) | 237 | 64.6 | 64 | 68.8 | 79 | 52.7 | 94 | 75.8 |
| Any other white background | 44 | 12.0 | 9 | 9.7 | 23 | 15.3 | 12 | 9.7 |
| Any Asian background | 72 | 19.6 | 16 | 17.2 | 42 | 28.0 | 14 | 11.3 |
| Any African or black background | 10 | 2.7 | 3 | 3.2 | 5 | 3.3 | 2 | 1.6 |
| Other | 4 | 1.1 | 1 | 1.1 | 1 | 0.7 | 2 | 1.6 |
| | | | | | | | | |
| HCW role | | | | | | | | |
| Nursing/midwifery | 181 | 49.3 | 57 | 61.3 | 69 | 46.0 | 55 | 44.4 |
| Allied health | 42 | 11.4 | 8 | 8.6 | 15 | 10.0 | 19 | 15.3 |
| Medical/dental | 73 | 19.9 | 11 | 11.8 | 33 | 22.0 | 29 | 23.4 |
| Administration | 24 | 6.5 | 6 | 6.5 | 9 | 6.0 | 9 | 7.3 |
| General support | 12 | 3.3 | 2 | 2.2 | 5 | 3.3 | 5 | 4.0 |
| Health care assistant | 32 | 8.7 | 8 | 8.6 | 19 | 12.7 | 5 | 4.0 |
| Other | 3 | 0.8 | 1 | 1.1 | 0 | 0.0 | 2 | 1.6 |

**Appendix 3E** COVID-19 related characteristics of participants who had a previously PCR-confirmed infection and their test results on Abbott SARS-CoV-2 IgG

| Parameter | All participants | | Test result Grayzone | | Positive | | Negative | |
|---|---|---|---|---|---|---|---|---|
| | *n* | % | *n* | % | *n* | % | *n* | % |
| Total | 367 | | 93 | | 150 | | 124 | |
| | | | | | | | | |
| Close contact with a case of COVID-19 | | | | | | | | |
| Yes | 211 | 57.5 | 56 | 60.2 | 90 | 60.0 | 65 | 52.4 |
| No | 154 | 42.0 | 37 | 39.8 | 58 | 38.7 | 59 | 47.6 |
| Unknown | 2 | 0.5 | 0 | 0.0 | 2 | 1.3 | 0 | 0.0 |
| | | | | | | | | |
| Main type of patient contact[a] | | | | | | | | |
| Daily contact with known/suspected COVID-19 patients | 88 | 24.0 | 22 | 23.7 | 40 | 26.7 | 26 | 21.0 |
| Daily contact with patients without COVID-19 | 218 | 59.4 | 56 | 60.2 | 88 | 58.7 | 74 | 59.7 |
| No patient contact | 61 | 16.6 | 15 | 16.1 | 22 | 14.7 | 24 | 19.4 |
| | | | | | | | | |
| Symptoms at time of previous PCR test | | | | | | | | |
| No symptoms | 7 | 1.9 | 1 | 1.1 | 3 | 2.0 | 3 | 2.4 |
| Had symptoms | 358 | 97.5 | 92 | 98.9 | 145 | 96.7 | 121 | 97.6 |
| Unknown | 2 | 0.5 | 0 | 0.0 | 2 | 1.3 | 0 | 0.0 |
| | | | | | | | | |
| Severity of symptoms | | | | | | | | |
| Minor symptoms | 151 | 41.1 | 38 | 43.2 | 51 | 35.2 | 62 | 51.2 |
| Significant symptoms | 178 | 48.5 | 48 | 54.5 | 75 | 51.7 | 55 | 45.5 |
| Severe symptoms (hospitalized) | 29 | 7.9 | 6 | 6.8 | 19 | 13.1 | 4 | 3.3 |
| | | | | | | | | |
| No. of mo since positive PCR test | | | | | | | | |
| Less than 1 | 8 | 2.2 | 1 | 1.1 | 6 | 4.0 | 1 | 0.8 |
| 1 | 8 | 2.2 | 0 | 0.0 | 8 | 5.3 | 0 | 0.0 |
| 2 | 4 | 1.1 | 0 | 0.0 | 4 | 2.7 | 0 | 0.0 |
| 3 | 1 | 0.3 | 0 | 0.0 | 1 | 0.7 | 0 | 0.0 |
| 4 | 12 | 3.3 | 2 | 2.2 | 10 | 6.7 | 0 | 0.0 |
| 5 | 85 | 23.2 | 28 | 30.1 | 34 | 22.7 | 23 | 18.5 |
| 6 | 222 | 60.5 | 54 | 58.1 | 77 | 51.3 | 91 | 73.4 |
| 7 | 25 | 6.8 | 8 | 8.6 | 8 | 5.3 | 9 | 7.3 |
| Unknown | 2 | 0.5 | 0 | 0.0 | 2 | 1.3 | 0 | 0.0 |
| | | | | | | | | |
| No. of mo since onset of symptoms[b] | | | | | | | | |
| Less than 1 | 5 | 1.5 | 1 | 1.1 | 3 | 2.2 | 1 | 0.9 |
| 1 | 8 | 2.4 | 0 | 0.0 | 8 | 5.8 | 0 | 0.0 |
| 2 | 3 | 0.9 | 0 | 0.0 | 3 | 2.2 | 0 | 0.0 |
| 3 | 1 | 0.3 | 0 | 0.0 | 1 | 0.7 | 0 | 0.0 |
| 4 | 12 | 3.5 | 2 | 2.3 | 10 | 7.2 | 0 | 0.0 |
| 5 | 78 | 22.9 | 25 | 28.4 | 32 | 23.2 | 21 | 18.4 |
| 6 | 210 | 61.8 | 52 | 59.1 | 73 | 52.9 | 85 | 74.6 |
| 7 | 23 | 6.8 | 8 | 9.1 | 8 | 5.8 | 7 | 6.1 |

[a]Participants were asked which one describes most of their current work.
[b]Excludes 27 participants who were not symptomatic at the time of their positive PCR test.

**Appendix 4** Factors associated with Abbott SARS-CoV-2 IgG seronegativity, among participants who had a previously PCR-confirmed infection, including those who had a grayzone result

| Parameter | All participants (n) | Negative result n | Negative result % | Unadjusted OR | Unadjusted 95% CI | Unadjusted P value[a] | Adjusted OR | Adjusted 95% CI | Adjusted P value[a] |
|---|---|---|---|---|---|---|---|---|---|
| Total | 367 | 124 | 33.8 | | | | | | |
| **Age group** | | | | | | | | | |
| 18–29 | 111 | 41 | 36.9 | *[b] | | | * | | |
| 30–39 | 108 | 43 | 39.8 | 1.13 | 0.66–1.95 | 0.662 | 0.96 | 0.51–1.8 | 0.893 |
| 40–49 | 79 | 17 | 21.5 | 0.47 | 0.24–0.89 | 0.024 | 0.44 | 0.21–0.92 | 0.031 |
| 50–59 | 50 | 19 | 38.0 | 1.05 | 0.52–2.08 | 0.897 | 1.05 | 0.47–2.36 | 0.897 |
| 60+ | 19 | 4 | 21.1 | 0.46 | 0.12–1.35 | 0.187 | 0.32 | 0.08–1.11 | 0.090 |
| **Sex** | | | | | | | | | |
| Female | 269 | 102 | 37.9 | * | | | * | | |
| **Male** | **98** | **22** | **22.4** | **0.47** | **0.28–0.81** | **0.006** | **0.38** | **0.21–0.68** | **0.001** |
| **Ethnicity** | | | | | | | | | |
| Irish (white) | 237 | 94 | 39.7 | * | | | * | | |
| Any other white background | 44 | 12 | 27.3 | 0.57 | 0.27–1.14 | 0.123 | 0.64 | 0.29–1.39 | 0.273 |
| Any Asian background | 72 | 14 | 19.4 | 0.37 | 0.19–0.68 | 0.002 | 0.41 | 0.19–0.81 | 0.012 |
| Any African/black background | 10 | 2 | 20.0 | 0.38 | 0.05–1.56 | 0.228 | 0.48 | 0.07–2.29 | 0.392 |
| Other | 4 | 2 | 50.0 | 1.52 | 0.18–12.9 | 0.677 | 1.53 | 0.15–15.7 | 0.705 |
| **Close contact with a case of COVID-19[c]** | | | | | | | | | |
| Yes | 211 | 65 | 30.8 | 0.72 | 0.46–1.11 | 0.147 | 0.67 | 0.4–1.12 | 0.127 |
| No | 154 | 59 | 38.3 | * | | | * | | |
| **Main type of patient contact[d]** | | | | | | | | | |
| No patient contact | 61 | 24 | 39.3 | * | | | * | | |
| Daily contact non-COVID-19 patients | 218 | 74 | 33.9 | 0.79 | 0.44–1.43 | 0.435 | 0.78 | 0.4–1.56 | 0.484 |
| Daily contact known or suspected COVID-19 patients | 88 | 26 | 29.5 | 0.65 | 0.21–0.32 | 0.214 | 0.84 | 0.38–1.89 | 0.679 |
| **Severity of symptoms[c]** | | | | | | | | | |
| No symptoms | 7 | 3 | 42.9 | 1.08 | 0.21–5.05 | 0.925 | 1.07 | 0.17–6.16 | 0.942 |
| Minor symptoms | 151 | 62 | 41.1 | * | | | * | | |
| Significant symptoms | 178 | 55 | 30.9 | 0.64 | 0.41–1.01 | 0.056 | 0.58 | 0.34–0.96 | 0.037 |
| Severe symptoms (hospitalized) | 29 | 4 | 13.8 | 0.23 | 0.06–0.63 | 0.009 | 0.26 | 0.07–0.76 | 0.023 |
| **HCW role** | | | | | | | | | |
| Administration | 24 | 9 | 37.5 | * | | | Not included in the model | | |
| Allied health care | 42 | 19 | 45.2 | 1.38 | 0.49–3.94 | 0.541 | | | |
| General support | 12 | 5 | 41.7 | 1.19 | 0.28–4.92 | 0.809 | | | |
| Health care assistant | 32 | 5 | 15.6 | 0.31 | 0.08–1.06 | 0.068 | | | |
| Medical/dental | 73 | 29 | 39.7 | 1.10 | 0.43–2.9 | 0.846 | | | |
| Nursing/midwifery | 181 | 55 | 30.4 | 0.73 | 0.31–1.83 | 0.481 | | | |
| Other | 3 | 2 | 66.7 | 3.33 | 0.28–78.0 | 0.353 | | | |
| **No. of mo since PCR positive test[c]** | | | | | | | | | |
| Less than 1 | 8 | 1 | 12.5 | NA | NA | NA | 0.23 | 0.01–1.6 | 0.204 |
| 1 | 8 | 0 | 0.0 | NA | NA | NA | NA | NA | NA |
| 2 | 4 | 0 | 0.0 | NA | NA | NA | NA | NA | NA |
| 3 | 1 | 0 | 0.0 | NA | NA | NA | NA | NA | NA |
| 4 | 12 | 0 | 0.0 | NA | NA | NA | NA | NA | NA |
| 5 | 85 | 23 | 27.1 | * | | | * | | |
| 6 | 222 | 91 | 41.0 | 1.87 | 1.09–3.29 | 0.025 | 1.95 | 1.07–3.64 | 0.032 |
| 7 | 25 | 9 | 36.0 | 1.52 | 0.57–3.87 | 0.389 | 2.07 | 0.72–5.9 | 0.172 |

[a]P values were calculated using the chi-square test, results for significant associations are highlighted in bold.
[b]*, reference category.
[c]Excludes two unknowns.
[d]Participants were asked which type of patient contact describes most of their current work (excludes 5 unknowns).

## ACKNOWLEDGMENTS

We acknowledge the study steering group who planned the study and critically evaluated the manuscript, the study team who coordinated the running of the study in each hospital, the hospital management at both sites for their support for the study, and the staff of both hospitals who participated. We would especially like to acknowledge the phlebotomy departments in each hospital for facilitating the sampling of almost 6,000 participants; the microbiology, virology, and biochemistry laboratories in each hospital for validating the assays and processing the samples on two different assays; the National Virus Reference Laboratory of Ireland for additional testing; and the human resources department for their help with denominator data.

This work was supported by the Irish Health Service Executive COVID-19 budget. Work by N.C. is funded partly by a Science Foundation Ireland (SFI) grant, grant code 20/SPP/3685.

We declare no conflicts of interest.

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
