## [Reviewer comments · Microbiology Spectrum]

Microbiology Spectrum

SARS-CoV-2 Antibody Testing in Healthcare Workers: a comparison of the clinical performance of three commercially available antibody assays

Niamh Allen, Melissa Brady, Antonio Isidro Carrion Martin, Llsa Domegan, Cathal Walsh, Elaine Houlihan, Colm Kerr, Lorraine Doherty, Joanne King, Martina Doheny, Damian Griffin, Maria Molloy, Jean Dunne, Vivion Crowley, Philip Holmes, Evan Keogh, Sean Naughton, Fiona O'Rourke, Martina Kelly, Yvonne Lynagh, Brendan Crowley, Cillian De Gascun, Paul Holder, Colm Bergin, Catherine Fleming, Una Ni Riain, and Niall Conlon

Corresponding Author(s): Niamh Allen, St.James's Hospital

Review Timeline:

Submission Date:	May 26, 2021
Editorial Decision:	July 19, 2021
Revision Received:	August 6, 2021
Editorial Decision:	August 15, 2021
Revision Received:	August 17, 2021
Accepted:	August 18, 2021

Editor: Yun Young Go

Reviewer(s): Disclosure of reviewer identity is with reference to reviewer comments included in decision letter(s). The following individuals involved in review of your submission have agreed to reveal their identity: Kamoru A ADEDOKUN (Reviewer #2)

Transaction Report:

DOI: <https://doi.org/10.1128/Spectrum.00391-21>

July 19, 2021

Dr. Niamh Allen
St. James's Hospital
Infectious Diseases
St. James's Street
Dublin Dublin 8
Ireland

Re: Spectrum00391-21 (SARS-CoV-2 Antibody Testing in Healthcare Workers: a comparison of the clinical performance of three commercially available antibody assays)

Dear Dr. Niamh Allen:

Thank you for submitting your manuscript to Microbiology Spectrum. When submitting the revised version of your paper, please provide (1) point-by-point responses to the issues raised by the reviewers as file type "Response to Reviewers," not in your cover letter, and (2) a PDF file that indicates the changes from the original submission (by highlighting or underlining the changes) as file type "Marked Up Manuscript - For Review Only". Please use this link to submit your revised manuscript - we strongly recommend that you submit your paper within the next 60 days or reach out to me. Detailed information on submitting your revised paper are below.

Link Not Available

Sincerely,

Yun Young Go

Journals Department
Editor comments:

Your manuscript has been considered by two reviewers recruited for their expertise in the field. As

their comments indicate, these individuals felt that your manuscript contained interesting observations but required modifications before publication could be considered acceptable. Importantly, the authors should clearly state whether the study focuses on the seroprevalence of COVID-19 in the Irish population or a comparative study among different assays as indicated in the title of the manuscript and modify the manuscript accordingly. In addition, the authors should consider reducing the length of the manuscript and present the data concisely and clearly. For example, tables 1 and 2 should be removed from the manuscript, while others (e.g., Tables 3, 4, 5, and 6) need improvements.

Reviewer comments:

Reviewer #1 (Comments for the Author):

This study compares the performance of a number of SARS-CoV-2 serological assays in the context of Irish healthcare workers. The assays compared are Roche and Abbott based on the Nucleocapsid protein tested on large numbers of samples (5,788) and the Wantai assay based on the Spike protein for a smaller number of samples (485). Overall, this is a well implemented study by a skilled and competent team with appropriate use of laboratory assays and statistical analyses.

- My primary comment is that the authors could be more definitive in their recommendation to select the Roche assay over the Abbott assay in the Irish context. For example, in the abstract:

"This may indicate that Roche is better suited than Abbott to population-based studies."

Given the different and consistent lines of evidence presented (serum samples from PCR positive individuals; comparison with a third assay), I think the authors can confidently drop the "may" from their statement.

Going further on this point, in the discussion the authors ask "Does the Abbott grayzone add anything?" This question is not given a direct answer, although all suggestions point towards no. My personal opinion is that the introduction of the grayzone is a belated attempt by Abbott to cover for poor performance of their assay.

- I personally find the manuscript a bit long with many tables. I appreciate the data is rich and there are many things that can be shown, but the authors should consider ways to reduce the length of the manuscript so that the points can be made more succinctly.

- In Table 4, I think it would be better to disaggregate 'negative' from 'grayzone'.

- Can Table 6 be extended to also include a comparison between Wantai and Abbott?

- The authors have provided a gold-standard evaluation of sensitivity by testing the serological assays on samples from individuals with previous PCR-confirmed SARS-CoV-2 infection. A limitation of the study is that they have not provided a comparable assessment of specificity by testing on pre-pandemic negative control samples. Could this limitation please be noted in the discussion.

- Figures 1 and 2 appear to visualize the same data in different ways. My understanding is that Figure 1 shows the results from each day, hence 0%, 50% and 100% are heavily over-represented. Figure 2 shows the same data in monthly intervals, but it's not clear how these intervals are

defined, e.g. what's the difference between "1< month" and "1"? What does "1" mean? I'll leave it up to the authors to decide how to proceed, but I would present a single figure like in Figure 2, but with very clearly defined intervals.

Reviewer #2 (Comments for the Author):

REVIEWER'S RESPONSE

I want to appreciate the efforts of the authors of this study by providing further possible info on varying assay results. It is my belief that this study will be a good reference point. However, worriedly, first, the authors should realize that certain things should be remembered while interpreting the test results. Having a high titer value does not always indicate an infection; similarly, having a low titer value is not necessarily associated with a low-grade infection or no infection - in fact, cross reactivity is becoming the bane of accurate diagnosis nowadays. Also, in attempting to answer a question of assay efficiency of this nature, it is important to understand the significant impact it may have on the product manufacturers, hence, every effort should be made to be clearly and highly disinterested - so as not to intentionally or unintentionally downgrade one firm for the other. On these aforementioned, I was curious to ask some questions of concern that were not addressed, despite a good study area of high importance.

Major Concerns:

□ How was reinfection taken care of or considered against misrepresentation of primary and secondary immune responses and their varying titres in the study, especially among the asymptomatics who never understood or knew they were positive until this study recruited them along?

□ The statement from L82-88 reads in part, "a study comparing four different serological assays showed a decline in the performance of the Abbott assay after 60 days, and a further decline after 80 days (12), whereas another study of the same assay showed a mean of 137 days to loss of positive antibody (13). Further data comparing antibody assays for the detection of antibodies to SARS-CoV-2 are needed to guide the most appropriate use of certain commercially available assays in any given situation". I think there should be a clause towards the conclusion based on this info, even there was an intra-assay nonspecificity, the patients' lack of detailed pathophysiological info such as the one (question above) querying possible reinfection or persistent infection among asymptomatics and the associated confusion in their immune responses would definitely affect Ab titre, assay performance and the outcome. So, the whole outcome shouldn't be based on assay inefficiency, unless otherwise CLEARLY treated. The reported, most efficient assay, might not be the case in the long run, this is to avoid misleading info. Clarity required.

□ A successful antibody titer result depends on the type of antibody being detected. For example, immunoglobulin M (IgM) appears in the blood between 2 - 4 weeks post-infection, whereas immunoglobulin G (IgG) takes around 4 - 6 weeks to achieve a detectable blood level. Thus, an appropriate test timing is important for a successful antibody titer. Definitely, an assay that tests for a single Ig (say, IgG) against the assay that tests for total antibodies (with different components that have varying titres/time) would emphatically have a conspicuous bias of measurement. This factor might have effect on figures 1 and 2, hence the general study conclusion. Clarity required.

□ Recall also, a statement made under discussion, Line 74-76, "Of the subset of participants who had previous PCR-confirmed COVID-19 infection, the Roche assay performed better in terms of

identifying these participants; 95% tested positive. Only half of these past infections were accurately identified by the Abbott assay". Even though, the report mentioned an important factor about selection bias that could undermine the conclusion, it is important to flag the first main idea stating that one assay, which tested for total Abs, was better than the other, that tested for only IgG, considering the difference in immunoglobulin differences, this would definitely pose effects on peaking and waning times after infection, and maybe reinfection. To add further, the statement on Lines 126-127 (Discussion) corroborates the importance of immunoglobulin peak-up time and other differences stating "within four weeks of antibody testing, 8/9 were correctly identified by the Abbott assay and 6/9 were identified by the Roche assay" obviously with higher percentage in this comparison. Clarity required.

□ Meanwhile, an antibody titer can only detect free antibodies in the blood; it is uncommon that any serological procedure allows detecting antibodies that are already bound to an antigen. Consequently, a person with a severe infection may test negative or have a low titer because of higher numbers of antigen-antibody complexes and lower numbers of free antibodies in the blood. Unfortunately, the present study, even though arose senses of eagerness in their seroprevalence study and possible indication of a better assay for improved testing of the virus, SARS-CoV-2, yet it is disappointing that the first point on the limitations was the central focus of the study from the background to the conclusion (and recommendation) while the authors maintained that this was not the core or thematic content of their study. I believe, sensitivity, specificity, and other accuracy-indicated studies would beneficially add to the body of diagnostic knowledge, especially currently that the world is in another mess of equivocal COVID-19 diagnostic outcomes - which have impacted human activities such as travel and economic effects, among others. The persisting and unresolved question in this statement ("Firstly, the study was not designed with the primary objective of comparing the serological assays" under limitation), however, is still ridiculously surprising and confusing!. Considering the study title, "SARS-CoV-2 Antibody Testing in Healthcare Workers: a comparison of the clinical performance of three commercially available antibody assays " and the short title: "Comparison of SARS-CoV-2 Antibody Assays", thus, it is obvious that, even though seroprevalence was a secondary audience target, the main idea is the assay comparison. The authors should clarify this further or do the necessary correction.

□ In certain conditions, a low titer may essentially indicate the efficient removal of infectious pathogens by the immune system. In contrast, a high titer may simply be due to the presence of residual antibodies from a previous infection, or unsuccessful attempts to form antigen-antibody complexes. In this current study, there is no any indication that this fact was treated. Of most worrisome is the comparison of an assay that uses IgG against the others with total Abs. Unsettling the sense of fairness, one may be worried that the authors failed to put this difference forth, but quick to pronounce the comparison results by adjudging with scientifically incongruous backgrounds favouring a manufacturer over the other. While I may be clear further here, the Wantai and Roche, both of which their results tallied, were based on total Abs. Segregationally, the Abbott, that uses IgG was quoted less in assay performance - whereas, the baseline difference was never accounted. I hope the authors can expatiate further on this green boundary.

Minor Corrections:

□ Some language corrections and misspellings need attention.

□ Words such as "SARS-CoV-2" and "COVID-19" in abstract and introduction should be first written in full at first mention.

□ Study design: The settings not mentioned, likewise, participants' info is hard to find here, and it is important. This section should give background info about the participants, sample size, and other characteristics of the participants. It is never in the result part where this was briefly mentioned.

Overall, this section is incomplete and short of information. "Both hospitals" as mentioned, what does this phrase mean? "All staff", what type of staffs are these? When was the study carried out? How were the participants recruited? What about ethical approval?

- Table 2 not clear -what do the columns 2 and 3 represent?
- Table 3 not communicating effectively. The columns have no indications of what they represent.
- L185 and L186 seem not correct with their percentages and corresponding numbers (n). And if true, then many figures require to be reviewed.
- Table 5, extra row on the table unnecessary. What does S/C rep?
- L205; "8.4% (n=485)" seems incorrect mathematically.
- L218; "Two-hundred and fifty-nine (71%)" seems not correct by calculation.
- Table 8; other columns signify no meanings except column 1. Also, the empty (1st) row on the table communicates nothing.
- Also, all the tables under the appendix similarly req attentions

Staff Comments:

Preparing Revision Guidelines

For complete guidelines on revision requirements, please see the Instructions to Authors at [link to page]. **Submissions of a paper that does not conform to Microbiology Spectrum guidelines will delay acceptance of your manuscript.**

Please return the manuscript within 60 days; if you cannot complete the modification within this time period, please contact me. If you do not wish to modify the manuscript and prefer to submit it to another journal, please notify me of your decision immediately so that the manuscript may be formally withdrawn from consideration by Microbiology Spectrum.

If you would like to submit an image for consideration as the Featured Image for an issue, please contact Spectrum staff.

If your manuscript is accepted for publication, you will be contacted separately about payment when the proofs are issued; please follow the instructions in that e-mail. Arrangements for payment

must be made before your article is published. For a complete list of **Publication Fees**, including supplemental material costs, please visit our website.

REVIEWER'S RESPONSE

I want to appreciate the efforts of the authors of this study by providing further possible info on varying assay results. It is my belief that this study will be a good reference point. However, worriedly, first, the authors should realize that certain things should be remembered while interpreting the test results. Having a high titer value does not always indicate an infection; similarly, having a low titer value is not necessarily associated with a low-grade infection or no infection - in fact, cross reactivity is becoming the bane of accurate diagnosis nowadays. Also, in attempting to answer a question of assay efficiency of this nature, it is important to understand the significant impact it may have on the product manufacturers, hence, every effort should be made to be clearly and highly disinterested - so as not to intentionally or unintentionally downgrade one firm for the other. On these aforementioned, I was curious to ask some questions of concern that were not addressed, despite a good study area of high importance.

Major Concerns:

- ❖ How was reinfection taken care of or considered against misrepresentation of primary and secondary immune responses and their varying titres in the study, especially among the asymptomatics who never understood or knew they were positive until this study recruited them along?
- ❖ The statement from L82-88 reads in part, *“a study comparing four different serological assays showed a decline in the performance of the Abbott assay after 60 days, and a further decline after 80 days (12), whereas another study of the same assay showed a mean of 137 days to loss of positive antibody (13). Further data comparing antibody assays for the detection of antibodies to SARS-CoV-2 are needed to guide the most appropriate use of certain commercially available assays in any given situation”*. I think there should be a clause towards the conclusion based on this info, even there was an intra-assay nonspecificity, the patients' lack of detailed pathophysiological info such as the one (question above) querying possible reinfection or persistent infection among asymptomatics and the associated confusion in their immune responses would definitely affect Ab titre, assay performance and the outcome. So, the whole outcome shouldn't be based on assay inefficiency, unless otherwise CLEARLY treated. The reported, most efficient assay, might not be the case in the long run, this is to avoid misleading info. Clarity required.
- ❖ A successful antibody titer result depends on the type of antibody being detected. For example, immunoglobulin M (IgM) appears in the blood between 2 – 4 weeks post-infection, whereas immunoglobulin G (IgG) takes around 4 – 6 weeks to achieve a detectable blood level. Thus, an appropriate test timing is important for a successful antibody titer. Definitely, an assay that tests for a single Ig (say, IgG) against the assay that tests for total antibodies (with different components that have varying titres/time) would emphatically have a conspicuous bias of measurement. This factor might have effect on figures 1 and 2, hence the general study conclusion. Clarity required.

- ❖ Recall also, a statement made under discussion, Line 74-76, “Of the subset of participants who had previous PCR-confirmed COVID-19 infection, the Roche assay performed better in terms of identifying these participants; 95% tested positive. Only half of these past infections were accurately identified by the Abbott assay”. Even though, the report mentioned an important factor about selection bias that could undermine the conclusion, it is important to flag the **first main idea** stating that one assay, which tested for total Abs, was better than the other, that tested for only IgG, considering the difference in immunoglobulin differences, this would definitely pose effects on peaking and waning times after infection, and maybe reinfection. To add further, the statement on Lines 126-127 (Discussion) corroborates the importance of immunoglobulin peak-up time and other differences stating “within four weeks of antibody testing, 8/9 were correctly identified by the Abbott assay and 6/9 were identified by the Roche assay” obviously with higher percentage in this comparison. Clarity required.
- ❖ Meanwhile, an antibody titer can only detect free antibodies in the blood; it is uncommon that any serological procedure allows detecting antibodies that are already bound to an antigen. Consequently, a person with a severe infection may test negative or have a low titer because of higher numbers of antigen-antibody complexes and lower numbers of free antibodies in the blood. Unfortunately, the present study, even though arose senses of eagerness in their seroprevalence study and possible indication of a better assay for improved testing of the virus, SARS-CoV-2, yet it is disappointing that the first point on the limitations was the central focus of the study from the background to the conclusion (and recommendation) while the authors maintained that this was not the core or thematic content of their study. I believe, sensitivity, specificity, and other accuracy-indicated studies would beneficially add to the body of diagnostic knowledge, especially currently that the world is in another mess of equivocal COVID-19 diagnostic outcomes - which have impacted human activities such as travel and economic effects, among others. The persisting and unresolved question in this statement (“Firstly, the study was not designed with the primary objective of comparing the serological assays” under **limitation**), however, is still ridiculously surprising and confusing!. Considering the study title, “SARS-CoV-2 Antibody Testing in Healthcare Workers: a comparison of the clinical performance of three commercially available antibody assays” and the short title: “Comparison of SARS-CoV-2 Antibody Assays”, thus, it is obvious that, even though seroprevalence was a secondary audience target, the main idea is the assay comparison. The authors should clarify this further or do the necessary correction.
- ❖ In certain conditions, a low titer may essentially indicate the efficient removal of infectious pathogens by the immune system. In contrast, a high titer may simply be due to the presence of residual antibodies from a previous infection, or unsuccessful attempts to form antigen-antibody complexes. In this current study, there is no any indication that this fact was treated. Of most worrisome is the comparison of an assay that uses IgG against the others with total Abs. Unsettling the sense of fairness, one may be worried that the authors failed to put this difference forth, but quick to pronounce the comparison results by adjudging with scientifically incongruous backgrounds favouring a manufacturer over the other. While I may be clear further here, the Wantai and Roche, both of which their

results tallied, were based on total Abs. Segregationally, the Abbott, that uses IgG was quoted less in assay performance - whereas, the baseline difference was never accounted. I hope the authors can expatiate further on this green boundary.

Minor Corrections:

- Some language corrections and misspellings need attention.
- Words such as “SARS-CoV-2” and “COVID-19” in abstract and introduction should be first written in full at first mention.
- **Study design:** The settings not mentioned, likewise, participants’ info is hard to find here, and it is important. This section should give background info about the participants, sample size, and other characteristics of the participants. It is never in the result part where this was briefly mentioned. Overall, this section is incomplete and short of information. “*Both hospitals*” as mentioned, what does this phrase mean? “*All staff*”, what type of staffs are these? When was the study carried out? How were the participants recruited? What about ethical approval?
- Table 2 not clear -what do the columns 2 and 3 represent?
- Table 3 not communicating effectively. The columns have no indications of what they represent.
- L185 and L186 seem not correct with their percentages and corresponding numbers (n). And if true, then many figures require to be reviewed.
- Table 5, extra row on the table unnecessary. What does S/C rep?
- L205; “8.4% (n=485)” seems incorrect mathematically.
- L218; “*Two-hundred and fifty-nine (71%)*” seems not correct by calculation.
- Table 8; other columns signify no meanings except column 1. Also, the empty (1st) row on the table communicates nothing.
- Also, all the tables under appendix similarly require attentions

Dear Yun Young Go

We would like to thank the editor, the editorial team and the reviewers for taking the time to read our manuscript. We are grateful to the reviewers for their helpful comments and hope that we have addressed them to their satisfaction.

Reviewer #1 (Comments for the Author):

This study compares the performance of a number of SARS-CoV-2 serological assays in the context of Irish healthcare workers. The assays compared are Roche and Abbott based on the Nucleocapsid protein tested on large numbers of samples (5,788) and the Wantai assay based on the Spike protein for a smaller number of samples (485). Overall, this is a well implemented study by a skilled and competent team with appropriate use of laboratory assays and statistical analyses.

Dear Reviewer,

Thank you very much for taking the time to review our work, and for your valuable comments, which we have addressed point-by-point below.

- My primary comment is that the authors could be more definitive in their recommendation to select the Roche assay over the Abbott assay in the Irish context. For example, in the abstract:

"This may indicate that Roche is better suited than Abbott to population-based studies."

Given the different and consistent lines of evidence presented (serum samples from PCR positive individuals; comparison with a third assay), I think the authors can confidently drop the "may" from their statement.

Dear Reviewer,

Thank you for your comment. We agree with you that that the "may" is probably too cautious in this instance, and have re-worded this as suggested, both in the abstract and in the discussion/ conclusion.

Going further on this point, in the discussion the authors ask "Does the Abbott grayzone add anything?" This question is not given a direct answer, although all suggestions point towards no. My personal opinion is that the introduction of the grayzone is a belated attempt by Abbott to cover for poor performance of their assay.

Thank you for this comment. Again, we agree, and were trying to word things conservatively. We have added a comment "so while the grayzone may add value for individual serology testing (by indicating a need for additional testing), we feel that it does not add value in the setting of population-based serological studies", to state more clearly our opinion that for this type of study, the grayzone does not add anything, and a better approach is to use a different test, as the sentence goes on to read "however the use of alternative assays with prolonged sensitivity over time would be superior if the primary purpose is to estimate infection ever". We have also shortened the comment on the grayzone in the conclusion to read instead, and more simply, "The Abbott grayzone did not add anything that would eliminate the need for additional testing on an alternative assay". Instead of "samples with a s/c index within the abbot grayzone still require additional testing on an alternative assay, and the numerical value within the grayzone S/C index range does not predict the result of confirmatory testing".

- I personally find the manuscript a bit long with many tables. I appreciate the data is rich and there

are many things that can be shown, but the authors should consider ways to reduce the length of the manuscript so that the points can be made more succinctly.

Thank you for this pertinent comment. We have cut down the text in all sections of the manuscript (500 words cut) while retaining the main message. We have also moved some tables (Tables 4, 5) and Figure 2A and 2B to the appendix that were perhaps not adding a lot of value in the main manuscript. We hope that you agree with the changes but would be very happy to receive further input on this, especially if the reviewer feels that anything pertinent has been cut.

- In Table 4, I think it would be better to disaggregate 'negative' from 'grayzone'.

Thank you to the reviewer for this suggestion. Unfortunately, it is not possible to separate the negative and grayzone in table 4 (now table 3), as we need an equal number of mutually exclusive result categories for each assay in order to test the kappa statistic. In other words, it's not possible to assess agreement between assays if we have two result categories (pos/neg) for one assay and three result categories (pos/gray/neg) for another assay. This is now noted in the methods.

- Can Table 6 be extended to also include a comparison between Wantai and Abbott?

Thank you to the reviewer for this suggestion. It would be possible to create a table showing comparison between Wantai and Abbott, however the study group previously discussed this and decided against it because of the sampling bias for testing on Wantai (those that were positive or grayzone on Abbott were selected for testing on Wantai). We feel that a table to this effect may overall be misleading. The Abbott grayzone category also makes interpretation of results difficult in table format, and overall we feel makes the results less clear. We are happy however to defer to the editor on preference here, as we do understand the reviewers' desire to see all comparison's side by side.

- The authors have provided a gold-standard evaluation of sensitivity by testing the serological assays on samples from individuals with previous PCR-confirmed SARS-CoV-2 infection. A limitation of the study is that they have not provided a comparable assessment of specificity by testing on pre-pandemic negative control samples. Could this limitation please be noted in the discussion.

Thank you to the reviewer for pointing out this very important limitation and we have added it to the manuscript. For the reviewers' information, an affiliated group did perform an assessment of specificity on pre-pandemic negative control samples in a small sideline study (on about 40 samples), which is pending publication in the Irish Medical journal. If this gets published before we finalise this manuscript we will reference this also.

- Figures 1 and 2 appear to visualize the same data in different ways. My understanding is that Figure 1 shows the results from each day, hence 0%, 50% and 100% are heavily over-represented. Figure 2 shows the same data in monthly intervals, but it's not clear how these intervals are defined, e.g. what's the difference between "1 < month" and "1"? What does "1" mean? I'll leave it up to the authors to decide how to proceed, but I would present a single figure like in Figure 2, but with very clearly defined intervals.

Thank you to the reviewer for this observation, on review the study team agree that the information could be presented in one figure only. We think Figure 1 is more visually impactful and tells the story more clearly, including highlighting more clearly that the majority of the infections were distant in time from the serology. We feel that Figure 2 is also important as it includes the Confidence Intervals, but perhaps could be moved to the appendix? We have moved it (Appendix 3A, 3B) to see how it flows, and we think it is an improvement but are happy to defer to the reviewer/editor on this. We have also amended the labelling of the x-axis and hope this improves clarity; 1, 2 3 etc referred to

number of months since positive PCR test, and perhaps the <1 at the beginning was confusing, so we have removed it. We have updated labelling of fig 2 as suggested (moved to Appendix 3A, 3B).

Reviewer #2 (Comments for the Author):

REVIEWER'S RESPONSE

I want to appreciate the efforts of the authors of this study by providing further possible info on varying assay results. It is my belief that this study will be a good reference point. However, worriedly, first, the authors should realize that certain things should be remembered while interpreting the test results. Having a high titer value does not always indicate an infection; similarly, having a low titer value is not necessarily associated with a low-grade infection or no infection - in fact, cross reactivity is becoming the bane of accurate diagnosis nowadays. Also, in attempting to answer a question of assay efficiency of this nature, it is important to understand the significant impact it may have on the product manufacturers, hence, every effort should be made to be clearly and highly disinterested - so as not to intentionally or unintentionally downgrade one firm for the other. On these aforementioned, I was curious to ask some questions of concern that were not addressed, despite a good study area of high importance.

Dear Reviewer,

Thank you very much for taking the time to review our work, and for your valuable comments, which we have addressed point-by-point below. Regarding the above, we absolutely agree that titre values do not necessarily correlate with infection/ no infection or severity of infection. Our investigation in this research regarding titre values was in order to determine if re-assigning COIs/S/C index of either assay would help with sensitivity in detecting previously confirmed infections. The answer was no, which we have clarified further in the text.

Please note that the line numbering starts again in the discussion, so there are duplicate line numbers. We highlight this to maximise clarity in our communication.

Major Concerns:

⊕How was reinfection taken care of or considered against misrepresentation of primary and secondary immune responses and their varying titres in the study, especially among the asymptomatics who never understood or knew they were positive until this study recruited them along?

Thank you for your this comment. We did not have any participants who reported known-reinfection by October 2020, when the study took place. It is possible that of those who reported infection eg. 6 months prior to serology, that they had been un-knowingly re-infected prior to serological testing, and that that might have contributed to the sustained serological response. However, we showed that Roche performed well on all those who reported only distant infection, and we feel it is unlikely that they had all been re-infected closer to the date of serological testing. Equally, Abbott performed worse with duration of time since infection, therefore re-infection is unlikely to explain this, as the opposite would be more likely; reinfection would be expected to enhance the ability of serological testing to pick up the infection.

Of those who had never known that they were infected, we did not use timelines for the interval between infection and serology in these patients, therefore we feel that even if they were infected twice, it would not have affected our analysis.

It is a point that is important to consider, and we have added this to limitations that we have no way of knowing if people were re-infected if they did not have 2 or more known PCR-confirmed infections. However, we do feel that it is unlikely that this was the case - at least for a significant number of participants - as evidence shows protection from re-infection for up to 9 months. We have added a reference to this effect also, as we agree that it is a pertinent point.

⌘The statement from L82-88 reads in part, "a study comparing four different serological assays showed a decline in the performance of the Abbott assay after 60 days, and a further decline after 80 days (12), whereas another study of the same assay showed a mean of 137 days to loss of positive antibody (13). Further data comparing antibody assays for the detection of antibodies to SARS-CoV-2 are needed to guide the most appropriate use of certain commercially available assays in any given situation". I think there should be a clause towards the conclusion based on this info, even there was an intra-assay nonspecificity, the patients' lack of detailed pathophysiological info such as the one (question above) querying possible reinfection or persistent infection among asymptomatics and the associated confusion in their immune responses would definitely affect Ab titre, assay performance and the outcome. So, the whole outcome shouldn't be based on assay inefficiency, unless otherwise CLEARLY treated. The reported, most efficient assay, might not be the case in the long run, this is to avoid misleading info. Clarity required.

Thank you for your comment. The quoted text here from the background is from studies that also have not commented on individual or aggregated individual pathophysiological differences, however these were studies large enough to hopefully have eliminated significant confounding due to individual pathophysiology. It is an important point to raise, and we had attempted to address this in limitations line 154 which reads "Other variables which would have been valuable to this study but were not available include participant co-morbidities and host determinants of immune response". We have now added "there may be individual pathophysiological factors that influence immune response differently depending on the antigenic target" as suggested and hope that this clarifies further that the authors are aware of the individual differences that may also be at play.

⌘A successful antibody titer result depends on the type of antibody being detected. For example, immunoglobulin M (IgM) appears in the blood between 2 - 4 weeks post-infection, whereas immunoglobulin G (IgG) takes around 4 - 6 weeks to achieve a detectable blood level. Thus, an appropriate test timing is important for a successful antibody titer. Definitely, an assay that tests for a single Ig (say, IgG) against the assay that tests for total antibodies (with different components that have varying titres/time) would emphatically have a conspicuous bias of measurement. This factor might have effect on figures 1 and 2, hence the general study conclusion. Clarity required.

Thank you to the reviewer for this comment. We agree that the timing of production of different antibodies is indeed paramount. In the discussion we feel that we have made this clear in lines 29-31 stating "the total antibody approach used in the Roche Elecsys system may result in improved and sustained sensitivity suitable for population seroprevalence studies." The preceding paragraph deals with the timing of production of IgG versus IgM. Furthermore, as the majority of our participants had distant infection (majority 6 months prior to antibody testing), we do not feel that the reason for the lack of IgG is antibody testing too early into course of infection, and therefore we feel that the general study conclusion remains robust. This is dealt with in results lines 230-233, and commented on further in the discussion. As well as this, figure 1 shows that the majority of PCR-confirmed infections had happened >150 days prior to serological sampling. We have also added a line to discussion to further clarify our position on this – line 63 now reads – "which, in agreement with our findings, suggest that total antibody assays are better positioned for population-based serological testing" and a line to the first paragraph of the conclusions stating "The performance of both total antibody assays (Roche and

Wantai) points towards the superiority of total antibody assays over IgG assays for serological studies.”

Re-infection, as mentioned above, may also impact this, and as outlined, we have added a comment on this in the limitations of the study, and reference to the evidence for 9 months of protection after natural infection, which makes re-infection in our cohort unlikely to have played a role.

We hope that this satisfactorily addresses the reviewers’ query.

⊞ Recall also, a statement made under discussion, Line 74-76, "Of the subset of participants who had previous PCR-confirmed COVID-19 infection, the Roche assay performed better in terms of identifying these participants; 95% tested positive. Only half of these past infections were accurately identified by the Abbott assay". Even though, the report mentioned an important factor about selection bias that could undermine the conclusion, it is important to flag the first main idea stating that one assay, which tested for total Abs, was better than the other, that tested for only IgG, considering the difference in immunoglobulin differences, this would definitely pose effects on peaking and waning times after infection, and maybe reinfection.

Thank you for this comment. We have strengthened our position on the preference of total Antibody assay over IgG only assay as discussed above, and also with clearer comments in the conclusion about the enhanced performance of the total antibody test.

We would also like to clarify that in this instance, and this comparison, there was no selection bias – all samples were tested on Roche and Abbott. The selection bias was for the Wantai.

To add further, the statement on Lines 126-127 (Discussion) corroborates the importance of immunoglobulin peak-up time and other differences stating "within four weeks of antibody testing, 8/9 were correctly identified by the Abbott assay and 6/9 were identified by the Roche assay" obviously with higher percentage in this comparison. Clarity required.

Thank you also for this observation, and this is why we included this comment, to highlight that different antigenic targets and immunological markers may be picked up at different times. However, the numbers here were far too small to draw significant conclusions about early pick up, and this was not the purpose of our research, so we do not feel we can comment further on these 9 cases.

⊞ Meanwhile, an antibody titer can only detect free antibodies in the blood; it is uncommon that any serological procedure allows detecting antibodies that are already bound to an antigen. Consequently, a person with a severe infection may test negative or have a low titer because of higher numbers of antigen-antibody complexes and lower numbers of free antibodies in the blood. Unfortunately, the present study, even though arose senses of eagerness in their seroprevalence study and possible indication of a better assay for improved testing of the virus, SARS-CoV-2, yet it is disappointing that the first point on the limitations was the central focus of the study from the background to the conclusion (and recommendation) while the authors maintained that this was not the core or thematic content of their study. I believe, sensitivity, specificity, and other accuracy-indicated studies would beneficially add to the body of diagnostic knowledge, especially currently that the world is in another mess of equivocal COVID-19 diagnostic outcomes - which have impacted human activities such as travel and economic effects, among others. The persisting and unresolved question in this statement ("Firstly, the study was not designed with the primary objective of comparing the serological assays" under limitation), however, is still ridiculously surprising and confusing!. Considering the study title, "SARS-CoV-2 Antibody Testing in Healthcare Workers: a comparison of the clinical performance of three commercially available antibody assays " and the short title: "Comparison of SARS-CoV-2 Antibody Assays", thus, it is obvious that, even though seroprevalence was a secondary audience

target, the main idea is the assay comparison. The authors should clarify this further or do the necessary correction.

Thank you for this comment. We have corrected the text at the start of limitations accordingly. The word study has been used in a confusing manner and we apologise for this; it has been used both to refer to the primary study (the PRECISE study) from which this data is taken, and to refer to this data analysis itself. We hope that we have sufficiently clarified this now.

⊞In certain conditions, a low titer may essentially indicate the efficient removal of infectious pathogens by the immune system. In contrast, a high titer may simply be due to the presence of residual antibodies from a previous infection, or unsuccessful attempts to form antigen-antibody complexes. In this current study, there is no any indication that this fact was treated. Of most worrisome is the comparison of an assay that uses IgG against the others with total Abs. Unsettling the sense of fairness, one may be worried that the authors failed to put this difference forth, but quick to pronounce the comparison results by adjudging with scientifically incongruous backgrounds favouring a manufacturer over the other. While I may be clear further here, the Wantai and Roche, both of which their results tallied, were based on total Abs. Segregationally, the Abbott, that uses IgG was quoted less in assay performance - whereas, the baseline difference was never accounted. I hope the authors can expatiate further on this green boundary.

Thank you very much for this comment, and we have clarified further in abstract, discussion, and conclusion that the declaration of potential superiority of one test over another in the context of seroprevalence studies only, is based on the antibody targets, and not on the manufacturer - the final line of the first paragraph of the conclusion now reads "The performance of both total antibody assays (Roche and Wantai) points towards the superiority of total antibody assays over IgG assays for serological studies". We agree with the need to be impartial and unbiased, and appreciate your comments on this.

Minor Corrections:

λSome language corrections and misspellings need attention.

λWords such as "SARS-CoV-2" and "COVID-19" in abstract and introduction should be first written in full at first mention.

Thank you very much, these terms are defined in the introduction. We will defer to the editor regarding preferences for the abstract, given the desire for brevity here, and the widespread use of these acromyms.

λStudy design: The settings not mentioned, likewise, participants' info is hard to find here, and it is important. This section should gives background info about the participants, sample size, and other characteristics of the participants. It is never in the result part where this was briefly mentioned. Overall, this section is incomplete and short of information. "Both hospitals" as mentioned, what does this phrase mean? "All staff", what type of staffs are these? When was the study carried out? How were the participants recruited? What about ethical approval?

Thank you for this very important point, and we apologise for this oversight. We had added some details to the study design section, and have referenced the main report of the principal study, where all further information can be found on study location, recruitment, and study participants. In the interest of brevity, we prefer to reference where this information is available, but if the editor would prefer us to include all of this information explicitly in the methods section we are happy to do so.

Regarding ethical approval, this is addressed after acknowledgements. We are happy to be advised by the editor regarding the placement of this if it would be preferred elsewhere, eg. In the methods section.

λTable 2 not clear -what do the columns 2 and 3 represent?

Column 2 is sensitivity and column 3 is specificity, and both are labelled accordingly. I see that when the PDF is printed out, for some reason the column labels come out blank. When viewed electronically they are labelled. Could the publishing team ensure no error if this manuscript is finally accepted.

λTable 3 not communicating effectively. The columns have no indications of what they represent.

The columns here are also labelled, same issue as above that has affected the reviewer

λL185 and L186 seem not correct with their percentages and corresponding numbers (n). And if true, then many figures require to be reviewed.

We have checked and verified that all of these numbers and percentages are correct, and have added the denominators for clarity.

λTable 5, extra row on the table unnecessary. What does S/C rep?

Same issue as above – these are not blank rows, they are labels for the columns, and are correct in the electronic version.

S/C is sample to calibrator (S/C) index) - defined in the methods section, line 113

λL205; "8.4% (n=485)" seems incorrect mathematically.

The number and proportion is correct, but on review we feel that the wording was confusing – thank you to the reviewer for noting this. The text has been re-worded and hopefully it is clearer now.

λL218; "Two-hundred and fifty-nine (71%)" seems not correct by calculation.

Thank you, we have double checked this and the numbers and proportions are correct; denominator has been added for clarity.

λTable 8; other columns signify no meanings except column 1. Also, the empty (1st) row on the table communicates nothing.

λAlso, all the tables under the appendix similarly req attentions

All same issue as raised above – all labelled correctly electronically but no appearing on PDF printout. We would appreciate the editorial team's review of this, thank you.

Staff Comments:

Preparing Revision Guidelines

For complete guidelines on revision requirements, please see the Instructions to Authors at [link to page]. **Submissions of a paper that does not conform to Microbiology Spectrum guidelines will delay acceptance of your manuscript.**

Please return the manuscript within 60 days; if you cannot complete the modification within this time period, please contact me. If you do not wish to modify the manuscript and prefer to submit it to another journal, please notify me of your decision immediately so that the manuscript may be formally withdrawn from consideration by Microbiology Spectrum.

If you would like to submit an image for consideration as the Featured Image for an issue, please contact Spectrum staff.

August 15, 2021

Dr. Niamh Allen
St. James's Hospital
Infectious Diseases
St. James's Street
Dublin Dublin 8
Ireland

Re: Spectrum00391-21R1 (SARS-CoV-2 Antibody Testing in Healthcare Workers: a comparison of the clinical performance of three commercially available antibody assays)

Dear Dr. Niamh Allen:

Thank you for submitting your manuscript to Microbiology Spectrum. As you will see your paper is very close to acceptance. Please modify the manuscript along the lines I have recommended. As these revisions are quite minor, I expect that you should be able to turn in the revised paper in less than 30 days, if not sooner. If your manuscript was reviewed, you will find the reviewers' comments below.

When submitting the revised version of your paper, please provide (1) point-by-point responses to the issues I raised in your cover letter, and (2) a PDF file that indicates the changes from the original submission (by highlighting or underlining the changes) as file type "Marked Up Manuscript - For Review Only". Please use this link to submit your revised manuscript. Detailed information on submitting your revised paper are below.

Link Not Available

Sincerely,

Yun Young Go

Editor comments:

- Present Figure 1 either as a single figure keeping only Figure 1A (both assays) OR Figures 1B (Abbott only) and 1C (Roche only).
- Please move "Ethical approval" information to "Study design" under Methods.

- Please check typographical mistakes throughout the manuscript.

Preparing Revision Guidelines

- point-by-point responses to the issues I raised in your cover letter
- Upload a compare copy of the manuscript (without figures) as a "Marked-Up Manuscript" file.
- Each figure must be uploaded as a separate file, and any multipanel figures must be assembled into one file.
- Manuscript: A .DOC version of the revised manuscript
- Figures: Editable, high-resolution, individual figure files are required at revision, TIFF or EPS files are preferred

For complete guidelines on revision requirements, please see the Instructions to Authors at [link to page]. **Submissions of a paper that does not conform to Microbiology Spectrum guidelines will delay acceptance of your manuscript.**

Please return the manuscript within 60 days; if you cannot complete the modification within this time period, please contact me. If you do not wish to modify the manuscript and prefer to submit it to another journal, please notify me of your decision immediately so that the manuscript may be formally withdrawn from consideration by Microbiology Spectrum.

If you would like to submit an image for consideration as the Featured Image for an issue, please contact Spectrum staff.

Dear Yun Young Go

We would like to thank the editor, the editorial team and the reviewers for taking the time to read our manuscript. We are grateful to the reviewers for their helpful comments and hope that we have addressed them to their satisfaction.

Reviewer #1 (Comments for the Author):

This study compares the performance of a number of SARS-CoV-2 serological assays in the context of Irish healthcare workers. The assays compared are Roche and Abbott based on the Nucleocapsid protein tested on large numbers of samples (5,788) and the Wantai assay based on the Spike protein for a smaller number of samples (485). Overall, this is a well implemented study by a skilled and competent team with appropriate use of laboratory assays and statistical analyses.

Dear Reviewer,

Thank you very much for taking the time to review our work, and for your valuable comments, which we have addressed point-by-point below.

- My primary comment is that the authors could be more definitive in their recommendation to select the Roche assay over the Abbott assay in the Irish context. For example, in the abstract:

"This may indicate that Roche is better suited than Abbott to population-based studies."

Given the different and consistent lines of evidence presented (serum samples from PCR positive individuals; comparison with a third assay), I think the authors can confidently drop the "may" from their statement.

Dear Reviewer,

Thank you for your comment. We agree with you that that the "may" is probably too cautious in this instance, and have re-worded this as suggested, both in the abstract and in the discussion/ conclusion.

Going further on this point, in the discussion the authors ask "Does the Abbott grayzone add anything?" This question is not given a direct answer, although all suggestions point towards no. My personal opinion is that the introduction of the grayzone is a belated attempt by Abbott to cover for poor performance of their assay.

Thank you for this comment. Again, we agree, and were trying to word things conservatively. We have added a comment "so while the grayzone may add value for individual serology testing (by indicating a need for additional testing), we feel that it does not add value in the setting of population-based serological studies", to state more clearly our opinion that for this type of study, the grayzone does not add anything, and a better approach is to use a different test, as the sentence goes on to read "however the use of alternative assays with prolonged sensitivity over time would be superior if the primary purpose is to estimate infection ever". We have also shortened the comment on the grayzone in the conclusion to read instead, and more simply, "The Abbott grayzone did not add anything that would eliminate the need for additional testing on an alternative assay". Instead of "samples with a s/c index within the abbot grayzone still require additional testing on an alternative assay, and the numerical value within the grayzone S/C index range does not predict the result of confirmatory testing".

- I personally find the manuscript a bit long with many tables. I appreciate the data is rich and there

are many things that can be shown, but the authors should consider ways to reduce the length of the manuscript so that the points can be made more succinctly.

Thank you for this pertinent comment. We have cut down the text in all sections of the manuscript (500 words cut) while retaining the main message. We have also moved some tables (Tables 4, 5) and Figure 2A and 2B to the appendix that were perhaps not adding a lot of value in the main manuscript. We hope that you agree with the changes but would be very happy to receive further input on this, especially if the reviewer feels that anything pertinent has been cut.

- In Table 4, I think it would be better to disaggregate 'negative' from 'grayzone'.

Thank you to the reviewer for this suggestion. Unfortunately, it is not possible to separate the negative and grayzone in table 4 (now table 3), as we need an equal number of mutually exclusive result categories for each assay in order to test the kappa statistic. In other words, it's not possible to assess agreement between assays if we have two result categories (pos/neg) for one assay and three result categories (pos/gray/neg) for another assay. This is now noted in the methods.

- Can Table 6 be extended to also include a comparison between Wantai and Abbott?

Thank you to the reviewer for this suggestion. It would be possible to create a table showing comparison between Wantai and Abbott, however the study group previously discussed this and decided against it because of the sampling bias for testing on Wantai (those that were positive or grayzone on Abbott were selected for testing on Wantai). We feel that a table to this effect may overall be misleading. The Abbott grayzone category also makes interpretation of results difficult in table format, and overall we feel makes the results less clear. We are happy however to defer to the editor on preference here, as we do understand the reviewers desire to see all comparison's side by side.

- The authors have provided a gold-standard evaluation of sensitivity by testing the serological assays on samples from individuals with previous PCR-confirmed SARS-CoV-2 infection. A limitation of the study is that they have not provided a comparable assessment of specificity by testing on pre-pandemic negative control samples. Could this limitation please be noted in the discussion.

Thank you to the reviewer for pointing out this very important limitation and we have added it to the manuscript. For the reviewers information, an affiliated group did perform an assessment of specificity on pre-pandemic negative control samples in a small sideline study (on about 40 samples), which is pending publication in the Irish Medical journal. If this gets published before we finalise this manuscript we will reference this also.

- Figures 1 and 2 appear to visualize the same data in different ways. My understanding is that Figure 1 shows the results from each day, hence 0%, 50% and 100% are heavily over-represented. Figure 2 shows the same data in monthly intervals, but it's not clear how these intervals are defined, e.g. what's the difference between "1 < month" and "1"? What does "1" mean? I'll leave it up to the authors to decide how to proceed, but I would present a single figure like in Figure 2, but with very clearly defined intervals.

Thank you to the reviewer for this observation, on review the study team agree that the information could be presented in one figure only. We think Figure 1 is more visually impactful and tells the story more clearly, including highlighting more clearly that the majority of the infections were distant in time from the serology. We feel that Figure 2 is also important as it includes the Confidence Intervals, but perhaps could be moved to the appendix? We have moved it (Appendix 3A, 3B) to see how it flows, and we think it is an improvement but are happy to defer to the reviewer/editor on this. We have also amended the labelling of the x-axis and hope this improves clarity; 1, 2 3 etc referred to

number of months since positive PCR test, and perhaps the <1 at the beginning was confusing, so we have removed it. We have updated labelling of fig 2 as suggested (moved to Appendix 3A, 3B).

Reviewer #2 (Comments for the Author):

REVIEWER'S RESPONSE

I want to appreciate the efforts of the authors of this study by providing further possible info on varying assay results. It is my belief that this study will be a good reference point. However, worriedly, first, the authors should realize that certain things should be remembered while interpreting the test results. Having a high titer value does not always indicate an infection; similarly, having a low titer value is not necessarily associated with a low-grade infection or no infection - in fact, cross reactivity is becoming the bane of accurate diagnosis nowadays. Also, in attempting to answer a question of assay efficiency of this nature, it is important to understand the significant impact it may have on the product manufacturers, hence, every effort should be made to be clearly and highly disinterested - so as not to intentionally or unintentionally downgrade one firm for the other. On these aforementioned, I was curious to ask some questions of concern that were not addressed, despite a good study area of high importance.

Dear Reviewer,

Thank you very much for taking the time to review our work, and for your valuable comments, which we have addressed point-by-point below. Regarding the above, we absolutely agree that titre values do not necessarily correlate with infection/ no infection or severity of infection. Our investigation in this research regarding titre values was in order to determine if re-assigning COIs/S/C index of either assay would help with sensitivity in detecting previously confirmed infections. The answer was no, which we have clarified further in the text.

Please note that the line numbering starts again in the discussion, so there are duplicate line numbers. We highlight this to maximise clarity in our communication.

Major Concerns:

⊕How was reinfection taken care of or considered against misrepresentation of primary and secondary immune responses and their varying titres in the study, especially among the asymptomatics who never understood or knew they were positive until this study recruited them along?

Thank you for your this comment. We did not have any participants who reported known-reinfection by October 2020, when the study took place. It is possible that of those who reported infection eg. 6 months prior to serology, that they had been un-knowingly re-infected prior to serological testing, and that that might have contributed to the sustained serological response. However, we showed that Roche performed well on all those who reported only distant infection, and we feel it is unlikely that they had all been re-infected closer to the date of serological testing. Equally, Abbott performed worse with duration of time since infection, therefore re-infection is unlikely to explain this, as the opposite would be more likely; reinfection would be expected to enhance the ability of serological testing to pick up the infection.

Of those who had never known that they were infected, we did not use timelines for the interval between infection and serology in these patients, therefore we feel that even if they were infected twice, it would not have affected our analysis.

It is a point that is important to consider, and we have added this to limitations that we have no way of knowing if people were re-infected if they did not have 2 or more known PCR-confirmed infections. However, we do feel that it is unlikely that this was the case - at least for a significant number of participants - as evidence shows protection from re-infection for up to 9 months. We have added a reference to this effect also, as we agree that it is a pertinent point.

⌘The statement from L82-88 reads in part, "a study comparing four different serological assays showed a decline in the performance of the Abbott assay after 60 days, and a further decline after 80 days (12), whereas another study of the same assay showed a mean of 137 days to loss of positive antibody (13). Further data comparing antibody assays for the detection of antibodies to SARS-CoV-2 are needed to guide the most appropriate use of certain commercially available assays in any given situation". I think there should be a clause towards the conclusion based on this info, even there was an intra-assay nonspecificity, the patients' lack of detailed pathophysiological info such as the one (question above) querying possible reinfection or persistent infection among asymptomatics and the associated confusion in their immune responses would definitely affect Ab titre, assay performance and the outcome. So, the whole outcome shouldn't be based on assay inefficiency, unless otherwise CLEARLY treated. The reported, most efficient assay, might not be the case in the long run, this is to avoid misleading info. Clarity required.

Thank you for your comment. The quoted text here from the background is from studies that also have not commented on individual or aggregated individual pathophysiological differences, however these were studies large enough to hopefully have eliminated significant confounding due to individual pathophysiology. It is an important point to raise, and we had attempted to address this in limitations line 154 which reads "Other variables which would have been valuable to this study but were not available include participant co-morbidities and host determinants of immune response". We have now added "there may be individual pathophysiological factors that influence immune response differently depending on the antigenic target" as suggested and hope that this clarifies further that the authors are aware of the individual differences that may also be at play.

⌘A successful antibody titer result depends on the type of antibody being detected. For example, immunoglobulin M (IgM) appears in the blood between 2 - 4 weeks post-infection, whereas immunoglobulin G (IgG) takes around 4 - 6 weeks to achieve a detectable blood level. Thus, an appropriate test timing is important for a successful antibody titer. Definitely, an assay that tests for a single Ig (say, IgG) against the assay that tests for total antibodies (with different components that have varying titres/time) would emphatically have a conspicuous bias of measurement. This factor might have effect on figures 1 and 2, hence the general study conclusion. Clarity required.

Thank you to the reviewer for this comment. We agree that the timing of production of different antibodies is indeed paramount. In the discussion we feel that we have made this clear in lines 29-31 stating "the total antibody approach used in the Roche Elecsys system may result in improved and sustained sensitivity suitable for population seroprevalence studies." The preceding paragraph deals with the timing of production of IgG versus IgM. Furthermore, as the majority of our participants had distant infection (majority 6 months prior to antibody testing), we do not feel that the reason for the lack of IgG is antibody testing too early into course of infection, and therefore we feel that the general study conclusion remains robust. This is dealt with in results lines 230-233, and commented on further in the discussion. As well as this, figure 1 shows that the majority of PCR-confirmed infections had happened >150 days prior to serological sampling. We have also added a line to discussion to further clarify our position on this – line 63 now reads – "which, in agreement with our findings, suggest that total antibody assays are better positioned for population-based serological testing" and a line to the first paragraph of the conclusions stating "The performance of both total antibody assays (Roche and

Wantai) points towards the superiority of total antibody assays over IgG assays for serological studies.”

Re-infection, as mentioned above, may also impact this, and as outlined, we have added a comment on this in the limitations of the study, and reference to the evidence for 9 months of protection after natural infection, which makes re-infection in our cohort unlikely to have played a role.

We hope that this satisfactorily addresses the reviewers’ query.

⊞Recall also, a statement made under discussion, Line 74-76, "Of the subset of participants who had previous PCR-confirmed COVID-19 infection, the Roche assay performed better in terms of identifying these participants; 95% tested positive. Only half of these past infections were accurately identified by the Abbott assay". Even though, the report mentioned an important factor about selection bias that could undermine the conclusion, it is important to flag the first main idea stating that one assay, which tested for total Abs, was better than the other, that tested for only IgG, considering the difference in immunoglobulin differences, this would definitely pose effects on peaking and waning times after infection, and maybe reinfection.

Thank you for this comment. We have strengthened our position on the preference of total Antibody assay over IgG only assay as discussed above, and also with clearer comments in the conclusion about the enhanced performance of the total antibody test.

We would also like to clarify that in this instance, and this comparison, there was no selection bias – all samples were tested on Roche and Abbott. The selection bias was for the Wantai.

To add further, the statement on Lines 126-127 (Discussion) corroborates the importance of immunoglobulin peak-up time and other differences stating "within four weeks of antibody testing, 8/9 were correctly identified by the Abbott assay and 6/9 were identified by the Roche assay" obviously with higher percentage in this comparison. Clarity required.

Thank you also for this observation, and this is why we included this comment, to highlight that different antigenic targets and immunological markers may be picked up at different times. However, the numbers here were far too small to draw significant conclusions about early pick up, and this was not the purpose of our research, so we do not feel we can comment further on these 9 cases.

⊞Meanwhile, an antibody titer can only detect free antibodies in the blood; it is uncommon that any serological procedure allows detecting antibodies that are already bound to an antigen. Consequently, a person with a severe infection may test negative or have a low titer because of higher numbers of antigen-antibody complexes and lower numbers of free antibodies in the blood. Unfortunately, the present study, even though arose senses of eagerness in their seroprevalence study and possible indication of a better assay for improved testing of the virus, SARS-CoV-2, yet it is disappointing that the first point on the limitations was the central focus of the study from the background to the conclusion (and recommendation) while the authors maintained that this was not the core or thematic content of their study. I believe, sensitivity, specificity, and other accuracy-indicated studies would beneficially add to the body of diagnostic knowledge, especially currently that the world is in another mess of equivocal COVID-19 diagnostic outcomes - which have impacted human activities such as travel and economic effects, among others. The persisting and unresolved question in this statement ("Firstly, the study was not designed with the primary objective of comparing the serological assays" under limitation), however, is still ridiculously surprising and confusing!. Considering the study title, "SARS-CoV-2 Antibody Testing in Healthcare Workers: a comparison of the clinical performance of three commercially available antibody assays " and the short title: "Comparison of SARS-CoV-2 Antibody Assays", thus, it is obvious that, even though seroprevalence was a secondary audience

target, the main idea is the assay comparison. The authors should clarify this further or do the necessary correction.

Thank you for this comment. We have corrected the text at the start of limitations accordingly. The word study has been used in a confusing manner and we apologise for this; it has been used both to refer to the primary study (the PRECISE study) from which this data is taken, and to refer to this data analysis itself. We hope that we have sufficiently clarified this now.

⊞In certain conditions, a low titer may essentially indicate the efficient removal of infectious pathogens by the immune system. In contrast, a high titer may simply be due to the presence of residual antibodies from a previous infection, or unsuccessful attempts to form antigen-antibody complexes. In this current study, there is no any indication that this fact was treated. Of most worrisome is the comparison of an assay that uses IgG against the others with total Abs. Unsettling the sense of fairness, one may be worried that the authors failed to put this difference forth, but quick to pronounce the comparison results by adjudging with scientifically incongruous backgrounds favouring a manufacturer over the other. While I may be clear further here, the Wantai and Roche, both of which their results tallied, were based on total Abs. Segregationally, the Abbott, that uses IgG was quoted less in assay performance - whereas, the baseline difference was never accounted. I hope the authors can expatiate further on this green boundary.

Thank you very much for this comment, and we have clarified further in abstract, discussion, and conclusion that the declaration of potential superiority of one test over another in the context of seroprevalence studies only, is based on the antibody targets, and not on the manufacturer - the final line of the first paragraph of the conclusion now reads "The performance of both total antibody assays (Roche and Wantai) points towards the superiority of total antibody assays over IgG assays for serological studies". We agree with the need to be impartial and unbiased, and appreciate your comments on this.

Minor Corrections:

λSome language corrections and misspellings need attention.

λWords such as "SARS-CoV-2" and "COVID-19" in abstract and introduction should be first written in full at first mention.

Thank you very much, these terms are defined in the introduction. We will defer to the editor regarding preferences for the abstract, given the desire for brevity here, and the widespread use of these acromyms.

λStudy design: The settings not mentioned, likewise, participants' info is hard to find here, and it is important. This section should gives background info about the participants, sample size, and other characteristics of the participants. It is never in the result part where this was briefly mentioned. Overall, this section is incomplete and short of information. "Both hospitals" as mentioned, what does this phrase mean? "All staff", what type of staffs are these? When was the study carried out? How were the participants recruited? What about ethical approval?

Thank you for this very important point, and we apologise for this oversight. We had added some details to the study design section, and have referenced the main report of the principal study, where all further information can be found on study location, recruitment, and study participants. In the interest of brevity, we prefer to reference where this information is available, but if the editor would prefer us to include all of this information explicitly in the methods section we are happy to do so.

Regarding ethical approval, this is addressed after acknowledgements. We are happy to be advised by the editor regarding the placement of this if it would be preferred elsewhere, eg. In the methods section.

λTable 2 not clear -what do the columns 2 and 3 represent?

Column 2 is sensitivity and column 3 is specificity, and both are labelled accordingly. I see that when the PDF is printed out, for some reason the column labels come out blank. When viewed electronically they are labelled. Could the publishing team ensure no error if this manuscript is finally accepted.

λTable 3 not communicating effectively. The columns have no indications of what they represent.

The columns here are also labelled, same issue as above that has affected the reviewer

λL185 and L186 seem not correct with their percentages and corresponding numbers (n). And if true, then many figures require to be reviewed.

We have checked and verified that all of these numbers and percentages are correct, and have added the denominators for clarity.

λTable 5, extra row on the table unnecessary. What does S/C rep?

Same issue as above – these are not blank rows, they are labels for the columns, and are correct in the electronic version.

S/C is sample to calibrator (S/C) index) - defined in the methods section, line 113

λL205; "8.4% (n=485)" seems incorrect mathematically.

The number and proportion is correct, but on review we feel that the wording was confusing – thank you to the reviewer for noting this. The text has been re-worded and hopefully it is clearer now.

λL218; "Two-hundred and fifty-nine (71%)" seems not correct by calculation.

Thank you, we have double checked this and the numbers and proportions are correct; denominator has been added for clarity.

λTable 8; other columns signify no meanings except column 1. Also, the empty (1st) row on the table communicates nothing.

λAlso, all the tables under the appendix similarly req attentions

All same issue as raised above – all labelled correctly electronically but no appearing on PDF printout. We would appreciate the editorial team's review of this, thank you.

Staff Comments:

Preparing Revision Guidelines

For complete guidelines on revision requirements, please see the Instructions to Authors at [link to page]. **Submissions of a paper that does not conform to Microbiology Spectrum guidelines will delay acceptance of your manuscript.**

Please return the manuscript within 60 days; if you cannot complete the modification within this time period, please contact me. If you do not wish to modify the manuscript and prefer to submit it to another journal, please notify me of your decision immediately so that the manuscript may be formally withdrawn from consideration by Microbiology Spectrum.

If you would like to submit an image for consideration as the Featured Image for an issue, please contact Spectrum staff.

August 18, 2021

Dr. Niamh Allen
St. James's Hospital
Infectious Diseases
St. James's Street
Dublin Dublin 8
Ireland

Re: Spectrum00391-21R2 (SARS-CoV-2 Antibody Testing in Healthcare Workers: a comparison of the clinical performance of three commercially available antibody assays)

Dear Dr. Niamh Allen:

Your manuscript has been accepted, and I am forwarding it to the ASM Journals Department for publication. You will be notified when your proofs are ready to be viewed.

Sincerely,

Yun Young Go
Editor, Microbiology Spectrum
